# Molybdenum isotopes unmask slab dehydration and melting beneath the Mariana arc

Hong-Yan Li [1,2,3 ✉], Rui-Peng Zhao[1,4], Jie Li[1,2], Yoshihiko Tamura [5], Christopher Spencer [6], Robert J. Stern [7], Jeffrey G. Ryan [8] & Yi-Gang Xu[1,2,3]

How serpentinites in the forearc mantle and subducted lithosphere become involved in enriching the subarc mantle source of arc magmas is controversial. Here we report molybdenum isotopes for primitive submarine lavas and serpentinites from active volcanoes and serpentinite mud volcanoes in the Mariana arc. These data, in combination with radiogenic isotopes and elemental ratios, allow development of a model whereby shallow, partially serpentinized and subducted forearc mantle transfers fluid and melt from the subducted slab into the subarc mantle. These entrained forearc mantle fragments are further metasomatized by slab fluids/melts derived from the dehydration of serpentinites in the subducted lithospheric slab. Multistage breakdown of serpentinites in the subduction channel ultimately releases fluids/melts that trigger Mariana volcanic front volcanism. Serpentinites dragged down from the forearc mantle are likely exhausted at >200 km depth, after which slab-derived serpentinites are responsible for generating slab melts.

[1] State Key Laboratory of Isotope Geochemistry, Guangzhou Institute of Geochemistry, Chinese Academy of Sciences, Guangzhou 510640, China. [2] CAS Center for Excellence in Deep Earth Science, Guangzhou 510640, China. [3] Southern Marine Science and Engineering Guangdong Laboratory (Guangzhou), Guangzhou 511458, China. [4] University of Chinese Academy of Sciences, Beijing 100049, China. [5] Research Institute for Marine Geodynamics (IMG), Japan Agency for Marine-Earth Science and Technology (JAMSTEC), Yokosuka 237-0061, Japan. [6] Department of Geological Sciences and Geological Engineering, Queen's University, Kingston, ON K7L 3N6, Canada. [7] Department of Geoscience, University of Texas at Dallas, Richardson, TX 75080, USA. [8] School of Geosciences, University of South Florida, Tampa, FL 33620, USA. ✉email: hongyanli@gig.ac.cn

The Izu-Bonin-Mariana (IBM) arc system stretches over 2800 km from near Tokyo, Japan to the south of Guam, USA, and is a typical intraoceanic arc system with negligible inputs from sub-arc oceanic crust to the lavas. The IBM system is an endmember non-accreting convergent margin with a thin sedimentary cover on the downgoing slab, which means that input and output fluxes in the subduction zone may be more confidently assessed[1]. The compositions of Mariana arc basalts indicate the involvement of two slab-derived components: an aqueous fluid enriched in fluid mobile elements (FMEs), characterized by high B/Nb, Ba/Nb, and Pb/Ce; and a hydrous melt, characterized by high Th/Nb and La/Sm but low B/Nb, Ba/Nb, and Pb/Ce[2–4]. The aqueous fluid component has been presumed to derive from the subducted altered mafic oceanic crust (AMOC) based on heavier boron isotopes and more radiogenic Nd isotopes, indicating a less sediment affected mantle source[2,4]. The hydrous melt component is from subducted sediments, given its high Th/Nb and less radiogenic Nd isotopes[2]. However, the conclusion that AMOC (average $^{87}Sr/^{86}Sr = \sim 0.7045$[5]) and marine sediment (average $^{87}Sr/^{86}Sr = \sim 0.710$[6]) act as respective sources for the aqueous fluid and hydrous melt is inconsistent with Mariana arc lavas having unradiogenic $^{87}Sr/^{86}Sr$ (0.7031–0.7041), and fresh mid-ocean ridge basalt (MORB)-like Pb isotopes[7–10], and mantle-like oxygen isotopic compositions[11]. Recent geochemical and geophysical observations suggest that the dehydration of serpentinites may play a more critical role in explaining these observed elemental and isotopic paradoxes. Two kinds of serpentinites may contribute to the mantle source of Mariana arc magmas. The first is serpentinite formed in the shallow forearc mantle at low temperatures (80–250 °C) above the shallow (<19 km) subducting slab[12–16]. Additions from forearc serpentinites to the sub-arc mantle or subduction channel can explain enrichments of FMEs and heavy boron isotopes (e.g., $\delta^{11}B$ from +4.5 to +12.0‰ in the Izu Arc[17,18]) in volcanic front basalts, as the slab experiences strong depletions of FMEs before reaching subarc depths[15,16,19–21]. The second serpentinite source is formed via seawater infiltrating oceanic mantle lithosphere along normal faults as the downgoing plate bends down to enter the trench[22]. Fluids resulting from serpentinite dehydration in the slab lithosphere[23,24] will heat up when rising toward the slab surface. These fluids increase extraction of FMEs from the crust, which could buffer the overall slab fluid/melt Sr–Pb isotope signature toward MORB values and also generate FME-enriched signatures[7,8,10,25,26].

While the importance of serpentinite subduction in the subduction zone chemical cycle is now widely acknowledged, it remains a formidable task to discriminate among these competing models. Which serpentinite sources are responsible, and how they work in the subduction zones are still unclear[27,28]. Molybdenum (Mo) is mobile in fluids, like Pb and Ba, but it can also be partially retained in rutile during slab dehydration or melting. Mo isotopic fractionation during slab devolatilization results in heavier isotopic signatures in the fluid phase, while the residual slab becomes isotopically lighter[8,29–33]. The materials of the subducting Pacific plate outboard of the Mariana arc have different Mo isotope compositions (expressed as $\delta^{98/95}Mo$, the ‰ deviation of $^{98}Mo/^{95}Mo$ from standard NIST SRM 3134). AMOC (−0.12‰ to +0.86‰, with a weighted average of +0.36‰[8]) and marine sediments (−1.87‰ to +0.11‰, with a weighted average of −0.31‰[8]) have $\delta^{98/95}Mo$ that are, respectively, higher and lower than the depleted mantle (DM; $\delta^{98/95}Mo = -0.21 \pm 0.02‰$[34]). Therefore, Mo isotopes may help distinguish among different slab components. This study reports new Mo elemental and isotopic data for well-characterized and very fresh lavas from Pagan and NW Rota-1 volcanoes in the Mariana arc, and serpentinite mud samples from the Asùt Tesoru (formerly Big Blue) Seamount in the Mariana forearc (Fig. 1 and Supplementary Fig. 1). Asùt Tesoru is an active mud volcano erupting serpentinite muds and entrained clasts from the

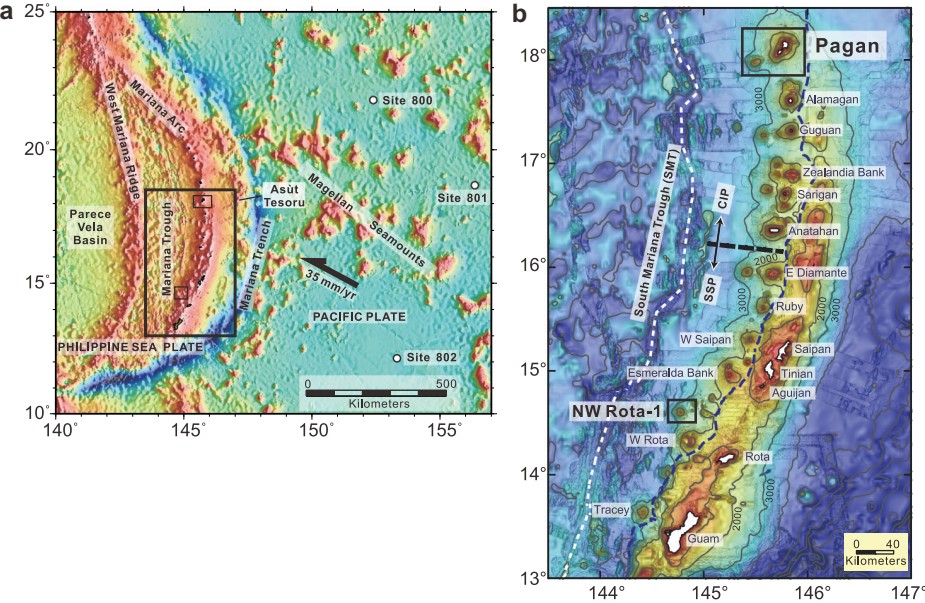

**Fig. 1 Regional map of the Mariana convergent plate margin with sampling locations. a** Regional map of the Mariana subduction system: the subducting Pacific Plate, Mariana Trench, Mariana Arc, Mariana Trough, West Mariana Ridge, and Parece Vela Basin. The Asùt Tesoru (Big Blue Seamount) mud volcano is located about 72 km west of the trench axis and 134 km east of the Pagan volcano and lies about 18 km above the downgoing Pacific plate[13]. Open circles denote Ocean Drilling Program drill Sites 800, 801, and 802. The rectangle shows the area enlarged in (**b**). **b** Map showing the southern Mariana arc, which includes from west to east the Mariana Trough (back-arc basin), the active Mariana arc, the old Mariana forearc (including the islands of Saipan, Tinian, Aguijan, Rota, and Guam), and the Mariana Trench. The white dashed line defines the Mariana Trough spreading ridge. The blue dashed line shows the boundary between the active Mariana arc to the west and the uplifted Mariana frontal arc, part of the Mariana forearc. The maps are from Tamura et al.[7]. SSP Southern Seamount Province, CIP Central Island Province.

subduction channel, located ~72 km west of the trench axis and 134 km east of Pagan Island. Asùt Tesoru is sited ~18 km above the downgoing plate[13]. Pagan lies along the volcanic front of the Central Island Province, while NW Rota-1 is a submarine volcano located ~40 km west of the volcanic front of the Southern Seamount Province. Based on the position of the slab beneath these centers[35], these two volcanoes represent different slab depths (~120–170 km vs. ~200–220 km; Supplementary Data 1), so the slab-derived phases should be different (more aqueous fluid beneath Pagan, more hydrous melt beneath NW Rota-1) as NW Rota-1 has a higher corresponding slab temperature than Pagan volcano. Both erupt little-fractionated high MgO basalts, which is unusual in arcs[7,36].

Here, we show that serpentinites formed in both the forearc mantle and the slab lithosphere are both important for generating subduction zone fluids/melts. Upper plate serpentinites transported down along the slab interface may be an important intermediate carrier for slab fluid and melt transfer in the subduction channel. Dehydration of serpentinites in the subducted lithospheric slab is critical for triggering melting at depth.

## Results

### Mo and $\delta^{98/95}$Mo of the volcanic lava and serpentinite mud samples.

Pagan and NW Rota-1 lavas reflect a wide range of magmatic fractionation, with MgO contents from 11.2 to 2.2 wt% (Fig. 2), allowing for evaluation of fractional crystallization effects on the Mo contents and isotopes of the lavas. The ranges in Mo contents among Pagan and NW Rota-1 samples at MgO > 7 wt% are similar (0.3–0.6 µg/g vs. 0.3–0.45 µg/g). However, the Pagan Southwestern (SW) Flank samples have higher Mo contents (1.0–1.9 µg/g) at <7 wt% MgO than do the NW Rota-1 samples (0.44–1.1 µg/g), suggesting different magmatic evolution processes (Fig. 2a). Negative correlations between Mo and MgO imply that Mo behaves as a moderately incompatible element during mantle melting and magmatic differentiation, similar to the light rare earth elements. Thus, Ce/Mo ratios can be used to document Mo enrichment in volcanic rocks or their mantle sources[8,30,31,33]. The SW Pagan samples point to complex magmatic evolution, including magma mixing, as indicated by reverse-zoned olivine and clinopyroxene phenocrysts[7]. The Dy/Yb of these samples decrease with decreasing MgO (Fig. 2b) and their Hf/Nd increase (Supplementary Fig. 2), suggesting fractional crystallization of amphibole from the evolved mixing component[37,38]. Ce/Mo ratios correlate positively with MgO in SW Pagan samples, while $\delta^{98/95}$Mo correlates inversely (Fig. 2c, d), suggesting that amphibole crystallization and/or mixing decreases Ce/Mo and increases $\delta^{98/95}$Mo with magmatic evolution. This fractionation trend is consistent with observations from differentiated arc lavas in other localities[39–41].

Dy/Yb and Hf/Nd show no clear correlation with MgO in the Eastern Flank or Summit samples of NW Rota-1 (Fig. 2b and Supplementary Fig. 2), indicating that the effects of amphibole crystallization are not significant. The East Knoll samples of NW Rota-1 vary little in MgO, making it difficult to evaluate fractional crystallization effects. However, they have similar Dy/Yb and Hf/Nd to other evolved NW Rota-1 samples (Fig. 2b and Supplementary Fig. 2), suggesting amphibole crystallization did not play a major role in their compositions. The evolved samples (<7 wt% MgO) from SW Pagan are excluded from the following discussion, to avoid the complexities introduced by fractional crystallization and/or mixing effects.

Pagan and NW Rota-1 samples that are unaffected by amphibole crystallization range in $\delta^{98/95}$Mo from −0.31‰ to +0.19 ‰ (Fig. 3), with values both higher and lower than that of DM (−0.21 ± 0.02‰[34,42]). This $\delta^{98/95}$Mo variation is much greater than has been observed in comparable Mariana arc lavas from the Central Island province (MgO: 2.7–6.1 wt%; $\delta^{98/95}$Mo = −0.17‰ to +0.15‰[8]) or in lavas from the Izu volcanic front (MgO:

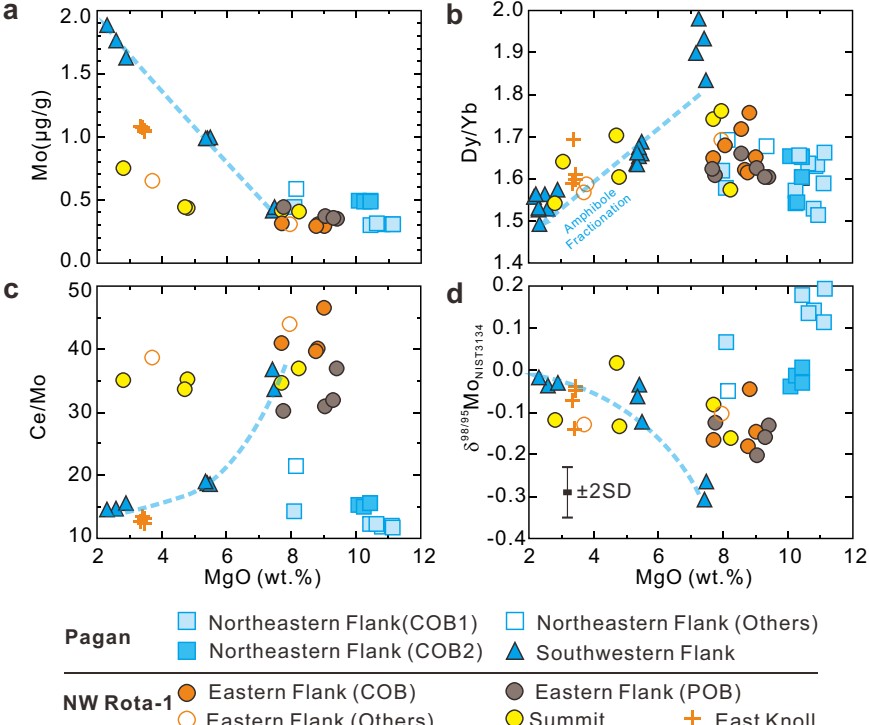

**Fig. 2 Chemical variations due to magmatic evolution to form Pagan and NW Rota-1 volcanic rocks.** Plots of **a** Mo, **b** Dy/Yb, **c** Ce/Mo, and **d** $\delta^{98/95}$Mo vs. MgO diagrams for the Pagan and NW Rota-1 samples. COB clinopyroxene-olivine basalt, POB plagioclase-olivine basalt, SD standard deviation, which is based on duplicate analyses of standard solution NIST SRM 3134, reference materials AGV-2 and W-2a, IAPSO seawater standard, and unknown samples.

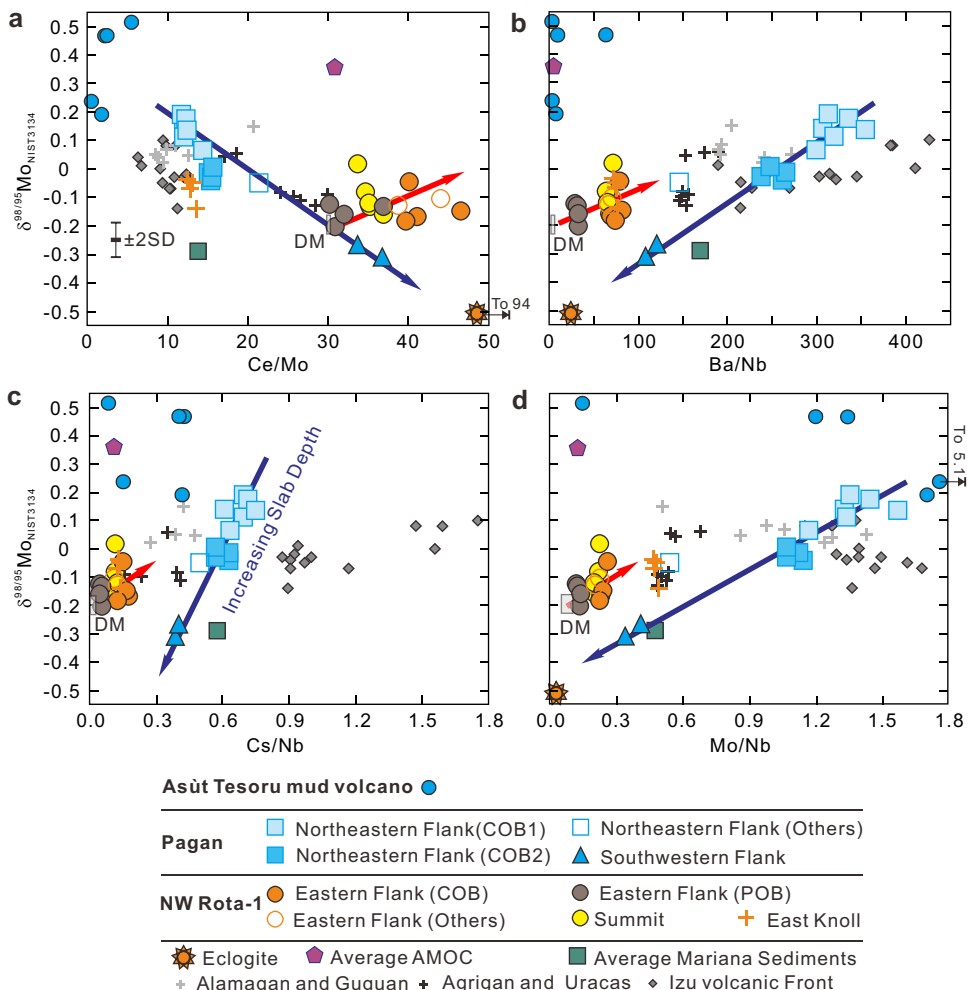

**Fig. 3 Mo isotope and trace element systematics of serpentinite mud and volcanic rock samples.** Plots of $\delta^{98/95}$Mo vs. **a** Ce/Mo, **b** Ba/Nb, **c** Cs/Nb, and **d** Mo/Nb for the Asùt Tesoru serpentinite mud samples and Pagan and NW Rota-1 volcanic rock samples unaffected by amphibole fractional crystallization. The Izu arc[33], other Mariana volcanic front volcanoes[8], average Mariana sediments[6,8], 801 C AMOC Super Composite[8,72] and average blueschist and eclogite data from the Raspas Complex and Cabo Ortegal Complex[32] are also plotted for comparison. The ranges for depleted mantle (DM) are estimated according to the studies of Bezard et al.[34], Gale et al.[44], Workman and Hart[73], and Salters and Stracke[43]. COB clinopyroxene-olivine basalt, POB plagioclase-olivine basalt, AMOC altered mafic oceanic crust, SD standard deviation. The blue arrows show compositions of magmas formed above increasing slab depth.

3.4–5.6 wt%; $\delta^{98/95}$Mo = −0.14‰ to +0.11‰[33]). The NW Rota-1 samples are lower in $\delta^{98/95}$Mo, from −0.20‰ to −0.02‰, and show a smaller $\delta^{98/95}$Mo range (0.18‰ ± 0.12‰) than either the Pagan samples (0.4‰ ± 0.12‰) or other IBM samples (0.32‰ ± 0.12‰). The Pagan and NW Rota-1 samples also show greater variation in Ce/Mo (11.8–46.5) than other Mariana arc (8.5–29.9) or Izu arc lavas (6.3–12.7; Fig. 3a), suggesting greater variability in magma source compositions. $\delta^{98/95}$Mo in the Pagan samples correlate with aqueous fluid proxies, e.g., Ba/Nb and Cs/ Nb ratios (Fig. 3b, c), consistent with an isotopically heavy Mo signature arising from slab dehydration. $\delta^{98/95}$Mo and Ce/Mo correlate inversely in the Pagan samples and show a much steeper trend than other Mariana and Izu volcanic front samples[8,33], suggesting a fluid-derived $\delta^{98/95}$Mo component that is much higher than +0.2‰.

All the NW Rota-1 samples save those from its East Knoll are similar in their trace element and Mo isotopic variations to the Pagan samples, in that Ba/Nb, Cs/Nb, and Mo/Nb are all higher than DM and correlate with $\delta^{98/95}$Mo (Fig. 3b–d), although they have somewhat higher Ce/Mo than DM (~31[43]). This observation

indicates that the slab beneath NW Rota-1 may already be significantly Mo depleted as a result of early dehydration. This deeper slab source may thus be similar to the eclogites from the Cabo Ortegal Complex in Spain, which experienced peak metamorphic conditions of >1.7 GPa and 650–670 °C, and have much lower Mo contents than average MORB (0.19 vs. 0.46 μg/ g[32,44]). Two of the SW Pagan samples have slightly higher Ce/Mo than DM and are much lower in $\delta^{98/95}$Mo than any of the NW Rota-1 samples. The East Knoll samples of NW Rota-1 have lower Ce/Mo than others from this center (Fig. 3a), possibly related to the greater Mo enrichment of their sources (Fig. 2a).

The three volcanic sample sites examined in this study reflect increasing distances from the trench, and therefore increasing depth to slab, in the following order: northeastern (NE) Pagan (124.5–129 km), SW Pagan (167–168 km), NW Rota-1 (198–223 km; Supplementary Data 1). A gradual reduction in fluid signatures with increasing slab depth (e.g., $\delta^{98/95}$Mo, Ce/Mo, and Ba/Nb) and increases in melt signatures (e.g., Hf/Nd; Fig. 4) indicate that the slab input correlates to slab temperature, evolving from a hydrous fluid-dominated phase to one that is

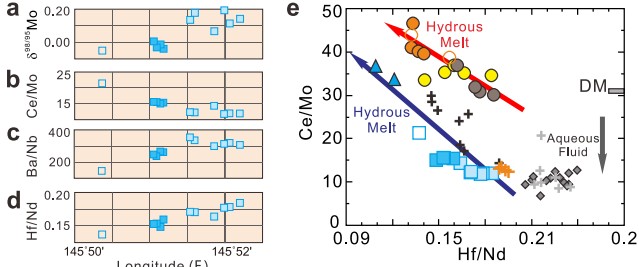

**Fig. 4 Chemical variations of Pagan volcanic rocks as a function of longitude and plot of Ce/Mo vs. Hf/Nd for Pagan and NW Rota-1 samples.** Plots of **a** $\delta^{98/95}$Mo, **b** Ce/Mo, **c** Ba/Nb, and **d** Hf/Nd vs. longitude for the NE Pagan samples, and **e** Ce/Mo vs. Hf/Nd for the Pagan and NW Rota-1 samples unaffected by amphibole fractional crystallization. The Izu arc and other Mariana volcanic front volcanoes data are also plotted for comparison. The ranges for depleted mantle (DM) are estimated according to the studies of Workman and Hart[73] and Salters and Stracke[43]. The blue arrows show compositions of magmas formed above increasing slab depth. The symbols and data source are as in Fig. 3.

melt dominated[3,21,45–47]. The samples from NE Pagan have a wide range of $\delta^{98/95}$Mo associated with more radiogenic Hf and Nd isotopes, but also with more radiogenic Sr and Pb isotopes as compared to the primitive NW-Rota-1 samples (Figs. 5 and 6). Radiogenic Hf and Nd isotopes are expected for melts derived from a depleted mantle source metasomatized by slab-derived aqueous fluids with negligible Hf–Nd contents. By contrast, the less radiogenic Hf and Nd of the primitive NW Rota-1 samples is expected for melts derived from a less depleted mantle, or from mantle metasomatized by a slab melt with contributions from subducting sediments. Mo isotopes reflect this fundamental difference, with higher $\delta^{98/95}$Mo in fluid-dominated NE Pagan samples and lower $\delta^{98/95}$Mo for slab melt-dominated NW Rota-1 samples. The Pagan samples show that $\delta^{98/95}$Mo correlates with Hf and Nd isotopes, albeit with limited Sr or Pb isotope variation. NW Rota-1 samples differ in that $\delta^{98/95}$Mo is generally positively correlated with $^{87}$Sr/$^{86}$Sr and $^{206}$Pb/$^{204}$Pb, but is not regularly correlated with Hf or Nd isotopes (Figs. 5 and 6).

The Asùt Tesoru serpentinite mud samples, which are associated with the highest Mo pore fluids among the analyzed serpentinite mud volcanoes in the Mariana forearc[48], have Mo contents from 0.1 to 1.3 μg/g and Ce/Mo from 0.5 to 5.5 (Fig. 3a; Supplementary Data 1). Their Mo contents are much higher and their Ce/Mo are much lower than DM (Mo = ~0.025 μg/g, Ce/Mo = ~31[43]). These samples show consistently high $\delta^{98/95}$Mo, from +0.19‰ to +0.52‰ (Fig. 3; Supplementary Data 1), much higher than DM. These data indicate mobilization of Mo from the slab under low P–T conditions in hydrous fluids (~18 km; ~250 °C[13,48]), and that Mo mobilized under these conditions is high $\delta^{98/95}$Mo. In Fig. 3, the Asùt Tesoru serpentinite muds appear to be consistent with the high $\delta^{98/95}$Mo endmember of the IBM arc lavas. However, they also have comparatively low Ba/Nb and Cs/Nb (Fig. 3).

**Tracking slab fluid and melt components for Pagan volcano.** The Mo isotope signatures of aqueous fluids from the subducting slab have been studied previously[8,30,33]. The wide variations in Ce/Mo and $\delta^{98/95}$Mo, and especially their correlations with aqueous fluid and hydrous melting proxies in this study, allow us to further discriminate among slab-derived fluid and melt signatures. The Mo isotopes of Pagan samples can be modeled as a two-component mixing array (Figs. 3–5). The high $\delta^{98/95}$Mo and low Ce/Mo signatures in NE Pagan samples largely reflect aqueous fluid contributions from the slab, given their correlations

with other fluid proxies such as Mo/Nb, Ba/Nb, and Cs/Nb[8,33]. This fluid component is similar to the Asùt Tesoru serpentinites in that they have high $\delta^{98/95}$Mo, low Ce/Mo, and high Mo/Nb. However, the high Cs/Nb and Ba/Nb signature of Mariana arc lavas (Fig. 3) are difficult to explain through serpentinite inputs alone, as Cs is only moderately (25 to 30%) while Ba is only slightly (<2%) mobile off the subducting slab at 10–40 km depths[15,16]. Further metasomatism of arc sources by slab fluids released at depths >40 km is necessary, under slab thermal conditions hot enough to mobilize Ba. While this fluid component has $\delta^{98/95}$Mo akin to AMOC (average $\delta^{98/95}$Mo = +0.36‰[8]), its unradiogenic $^{87}$Sr/$^{86}$Sr (~0.7035) and unaltered MORB-like Pb isotopes (Δ8/4 > −4; Fig. 6c) rule out a dominant role for AMOC as a source[7,8,10]. These isotopic observations have recently been accounted for via inputs from the breakdown of serpentinite in the slab lithosphere, which releases fluids that percolate through and equilibrate with overlying oceanic crust, buffering its Pb-Sr isotopes toward that of MORB[7,8,10]. Isotopic fractionation between the fluid and residual slab material can generate the $\delta^{98/95}$Mo signature of the fluid component evidenced in samples from NE Pagan[8].

The second mixing endmember has low $\delta^{98/95}$Mo (~−0.31‰) with higher Ce/Mo, low Ba/Nb, Cs/Nb, and Mo/Nb, and less radiogenic Hf–Nd isotopes (Figs. 3 and 5). This endmember appears most consistent with a slab melt component. The positive correlation between $\delta^{98/95}$Mo and εHf suggests a relationship between the addition of aqueous fluids and compositional heterogeneity of the sub-arc mantle, i.e., more aqueous fluid flux must be added to a more depleted and refractory sub-arc mantle for melting to occur[33]. However, the $\delta^{98/95}$Mo of primitive Pagan samples also correlate with Hf/Nd, a proxy for hydrous melt inputs from the slab[49–51]. This slab melt component has an Nd–Hf isotope signature akin to the Indian mantle domain as opposed to subducted Pacific crust (Fig. 5c). One can resolve these trace element and isotopic features via the addition of ≈10 wt% sediments to the melted slab[50,52–55] (Fig. 5c and Supplementary Fig. 3). The amount of subducted sediment contribution is limited by the samples' low $^{87}$Sr/$^{86}$Sr and $^{206}$Pb/$^{204}$Pb, which as noted above are consistent with an evolved serpentinite fluid contribution in which Sr–Pb isotopes are buffered toward MORB-like values[10,25,26] (Fig. 6).

Therefore, both the signatures of aqueous slab fluids and hydrous slab melts may be derived in different proportions from the different slab constituents (wedge and slab serpentinite, basalt, and sediments), and increasing slab temperatures with depth should control the generation of fluids and melts. How slab fluids and melts are injected into the sources of arc melting is still unclear, although their geochemical signatures are readily discriminated[7,21,25,26]. The most straightforward model is one in which Mo–Nd–Hf elemental and isotopic variations of Pagan magmas are controlled by the slab surface temperature, i.e., slab inputs richer in fluids gradually evolve toward a more hydrous melt signature with increasing slab depth. However, NE Pagan samples show clear differences in slab input with very little corresponding slab depth variation, 124.5–129 km[35] (Supplementary Data 1). Significant slab temperature variation over such a narrow depth range is unlikely[56]. Based on major and trace element and radiogenic isotope results, Tamura et al.[7] proposed that aqueous fluid and hydrous melt components unmix from an originally homogeneous supercritical fluid in or above the subducting slab below Pagan volcano. The geochemical diversity of the basalts erupted at NE and SW Pagan becomes possible if these two unmixed components are added separately to the mantle beneath the Pagan volcano. However, this explanation fails to explain the observed Mo isotope variations, including values both

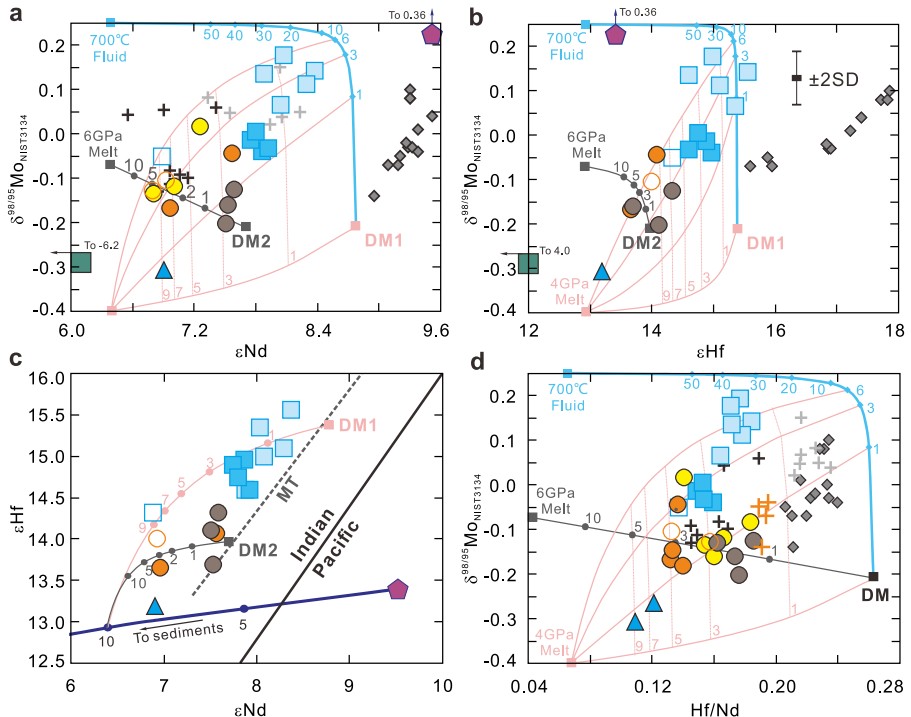

**Fig. 5 Mo, Nd, and Hf isotopes and Hf/Nd of Pagan and NW Rota-1 volcanic rocks.** Plots of **a** $\delta^{98/95}$Mo vs. $\varepsilon$Nd, **b** $\delta^{98/95}$Mo vs. $\varepsilon$Hf, **c** $\varepsilon$Hf vs. $\varepsilon$Nd, and **d** $\delta^{98/95}$Mo vs. Hf/Nd for the Pagan and NW Rota-1 samples unaffected by amphibole fractional crystallization, illustrating the slab dehydration/melting process. The Izu, other Mariana volcanic front, average Mariana sediments, and 801 C AMOC Super Composite data are also plotted for comparison. The depleted mantle 1 (DM1), DM2, 700 °C slab fluid, 4 GPa slab melt, and 6 GPa slab melt compositions are listed in Supplementary Table 1. The numbers on the mixing curves between different compositions represent the mass percentage of the slab fluid/melt. The solid black line in (**c**) is the boundary of Hf–Nd isotopic composition between the Indian and Pacific Ocean mantle domains[50,54]. The Hf–Nd isotopic trend of the Mariana Trough (MT) mantle is according to Woodhead et al.[53]. SD standard deviation. The symbols and data source are as in Fig. 3.

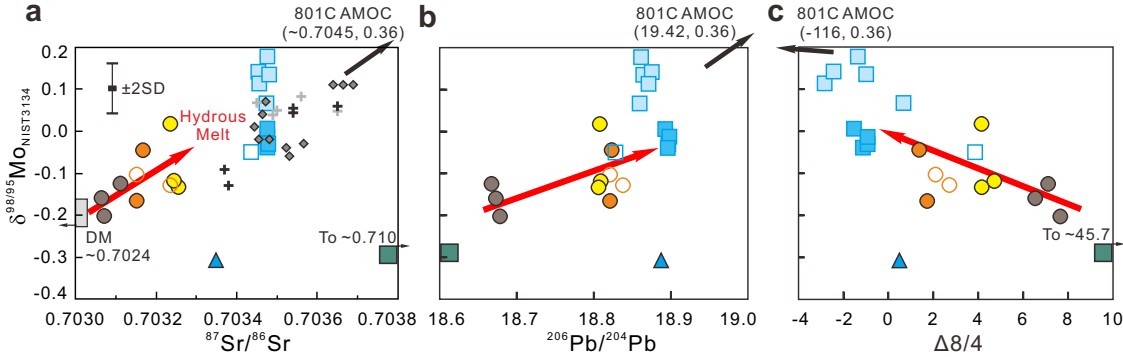

**Fig. 6 Mo, Sr, and Pb isotopes of Pagan and NW Rota-1 volcanic rocks.** Plots of **a** $\delta^{98/95}$Mo vs. $^{87}$Sr/$^{86}$Sr, **b** $\delta^{98/95}$Mo vs. $^{206}$Pb/$^{204}$Pb and **c** $\delta^{98/95}$Mo vs. $\Delta8/4$ for the Pagan and NW Rota-1 samples unaffected by amphibole fractional crystallization. $\Delta8/4$ is the displacement from the Northern Hemisphere Reference Line in $^{208}$Pb/$^{204}$Pb–$^{206}$Pb/$^{204}$Pb space[74]. The Sr-Mo isotopes of the depleted mantle (DM) are estimated according to the study on Pacific–Antarctic ridge basalts[34]. The Izu, other Mariana volcanic front, and average Mariana sediments data are also plotted for comparison. The ranges for the average 801 C AMOC[5,8,52] run off of the diagrams. The symbols and data source are as in Fig. 3. SD standard deviation, AMOC altered mafic oceanic crust.

higher and lower than DM, and a decrease in $\delta^{98/95}$Mo, from +0.19‰ to −0.31‰, correlated with the slab depths beneath the knolls (Fig. 4), as the two unmixed components should have similar Mo isotopes.

We conducted quantitative mixing calculations based on Mo–Hf–Nd elemental and isotopic systematics to address this problem. Detailed input parameters and calculated slab components for the Mariana arc lavas are listed in Supplementary Table 1. From $\varepsilon$Hf, $\varepsilon$Nd, and Hf/Nd systematics, the slab surface is estimated to be comprised of AMOC and sediments in a ratio of 9:1, and the

ambient mantle has $\varepsilon$Nd between 7.7 and 9.7 (Fig. 5c and Supplementary Fig. 3). Elemental abundances in the shallow slab fluid (700 °C) and slab melt (900 °C, 4 GPa) are calculated according to the liquid-eclogite partition coefficients of Kessel et al.[49], assuming modal batch dehydration/melting at $F = 2\%$ and $F = 10\%$, respectively. The shallow slab fluid $\delta^{98/95}$Mo is estimated to be the same as that for the Izu arc (+0.25‰[33]), while its Mo content is constrained by the uniform Ba/Mo = 230 for NE Pagan samples and the calculated Ba content for the fluid. The oceanic crust during melting is inferred to have the same average Ce/Mo and $\delta^{98/95}$Mo as those of

blueschists and eclogites from the Raspas Complex and Cabo Ortegal Complex[32]. Slab melt Mo is calculated using the liquid-mineral Mo partition coefficients of Chen et al.[32] and Adam and Green[57], assuming 2 wt% rutile in the slab, and a clinopyroxene:garnet ratio of 7:3. The Mo isotope fractionation factor is calculated according to the experimental result of $\Delta^{98/95}Mo_{melt-rutile} = 0.33 \pm 0.06$‰ at 1175 °C between the melt and the residual rutile[32]. Our calculations show that a slab melt at 4 GPa has $\delta^{98/95}Mo$ of −0.4‰, slightly lower than the lowest SW Pagan sample (−0.31‰), Ce/Mo of ~43, slightly higher than the highest SW Pagan sample (~37), and Hf/Nd of ~0.07, similar to that estimated for the slab melt component (Fig. 5d and Supplementary Fig. 3). The results also indicate a deep lithosphere fluid flux is not necessary for slab melting beneath SW Pagan, because it would be difficult to get the high Ce/Mo signature of the melt. The Sr–Pb isotopes of the melted slab may have been buffered toward the MORB values as a result of long-term early lithosphere fluid percolation at shallow depth.

Our calculations demonstrate that solely low-temperature slab fluid inputs cannot explain the low Hf/Nd characteristics of NE Pagan samples. An upper-mantle source would require ~30–40 wt% input of a 700 °C slab fluid or ~20 wt% input of an 800 °C slab fluid to lower its Hf/Nd from DM-like values, as Hf and Nd are so poorly mobilized in fluids, even at 700–800 °C (Fig. 5d). The constraints from Hf/Nd are independent of any estimation for Mo isotopes in a shallow slab fluid. Such a high percentage of slab addition is likely impossible, especially considering that the NE Pagan has a corresponding slab depth >120 km. Although the role of mixing slab components into the mantle wedge or mixing of magmas in the plumbing system cannot be fully ruled out[58], this does not explain the Th/Nb versus Pb/Ce variation in the Pagan samples (Supplementary Fig. 4). A more plausible explanation is that the source material for the Pagan samples was first metasomatized by 1–6% aqueous fluid with high $\delta^{98/95}Mo$, and is then fluxed by 1–9% hydrous melt with low $\delta^{98/95}Mo$, Hf/Nd, εNd, and εHf (Fig. 5 and Supplementary Fig. 5). Our calculations indicate that contributions from shallow aqueous fluids are smaller, while inputs of slab melt are larger for Pagan samples, consistent with the changes expected with increasing slab depth. The fluid signature apparently develops in the source before the melt signature in the Pagan mantle source, so it is most likely inherited from forearc serpentinites (>40 km depth) dragged down along the slab/mantle interface.

**Characterizing slab fluid and melt components for NW Rota-1 volcano.** All the NW Rota-1 samples aside from those from East Knoll show evidence for inputs from a slab source with higher Mo/Nb, Ba/Nb, and Cs/Nb than DM (Fig. 3). These samples are also characterized by unusually high Ce/Mo, which increases with decreasing Hf/Nd (Fig. 4e). Thus, we interpret the subduction input that fluxed the NW Rota-1 magma source to have been a hydrous melt. However, direct melting of a deeply subducted slab with a Mo isotopic composition like the eclogites from the Raspas Complex and Cabo Ortegal Complex cannot generate the Mo isotopic signature of NW Rota-1 samples (Fig. 5), so $\delta^{98/95}Mo$ must be higher at the slab surface at 200–220 km beneath NW Rota-1 than that at ~170 km beneath SW Pagan. Fluids from serpentinite decomposition in the slab lithosphere, equilibrated with MORB-like overlying crust, are in theory the only source for heavy $\delta^{98/95}Mo$ at such depths. The contrasting phenomena of $\delta^{98/95}Mo$ decreases and increases for shallow[32] (<65 km) and deep (>200 km) subduction, respectively, indicate that the slab lithosphere fluid plays different roles in trigging dehydration/melting of the slab crust. At >200 km, both of the upper and lower slab crust may experience Mo loss as a result of early melting and fluid percolation, respectively, at shallow

depth. We have calculated the $\delta^{98/95}Mo$ and trace element compositions of a slab at 200 km (6 GPa). The lower ocean crust is assumed to have experienced early fluid percolation (F = 2%) and Mo loss at shallow depth with $\delta^{98/95}Mo$ of ~ −0.44‰, while the upper ocean crust is assumed to be the residue of melting (F = 10%) at 130 km (4 GPa) with $\delta^{98/95}Mo$ of ~−0.9‰. Our calculations show that melting (F = 10%) of the upper oceanic crust (original $\delta^{98/95}Mo = -0.9$‰) with the addition of 2 wt% slab lithosphere fluid ($\delta^{98/95}Mo = +0.06$‰) equilibrated with the lower crust can produce a slab melt with $\delta^{98/95}Mo$ of ~−0.07‰ and Ce/Mo of ~51 (Supplementary Fig. 5), similar to the components for NW Rota-1 samples and consistent with the hypothesis that slab lithosphere serpentinite breakdown can generate the fluids that trigger melting in deeply subducted slabs[7,8,10,25,26]. This interpretation is supported by the lower $^{87}Sr/^{86}Sr$ in primitive NW Rota-1 lavas relative to Pagan (Fig. 6a). The low Ce/Mo and high Hf/Nd signatures of the East Knoll samples indicate a mantle source with a strong hydrous fluid signature, assuming these samples have not experienced significant fractional crystallization of amphibole (Figs. 4e and 5d).

The role of sub-arc mantle heterogeneity can be characterized through correlations between Nd–Hf isotopes and Hf/Nd ratios (Supplementary Fig. 3), which suggest that the ambient mantle beneath NW Rota-1 may have lower εNd and εHf than the mantle beneath Pagan, resulting in two trends on $\delta^{98/95}Mo$ vs. εNd and εHf, and εHf vs. εNd diagrams (Fig. 5). That the back-arc mantle source may be more fertile than that beneath the volcanic front has also been observed in the Izu-Bonin arc[46]. Per constraints from εHf, εNd, and Hf/Nd, the NW Rota-1 mantle source would appear to include 0–5% slab melt, assuming its ambient mantle is more enriched than that of Pagan volcano (Fig. 5).

## Discussion

Serpentinite breakdown has long been believed to play a crucial role in element transfer in subduction zones[8,15,16]. However, whether this serpentinite formed in the downgoing oceanic lithosphere or in the fore-arc mantle has been difficult to resolve[27,28]. Mo isotopes may help distinguish the respective roles of these two serpentinite sources.

Mo isotopes in NW Rota-1 lavas reflect a mantle source fluxed by hydrous melts generated via serpentinite breakdown in the subducting oceanic lithosphere, which triggered melting in the overlying subducted oceanic crust and sediments[7]. This mechanism fails to explain observed Mo isotope variations in the Pagan lavas (Figs. 3 and 5). Mo isotope variations and their correlations with elemental proxies for aqueous fluid and hydrous melt require that the source material for Pagan melts was first metasomatized by an aqueous fluid that raised its $\delta^{98/95}Mo$ and lowered its Ce/Mo without affecting its Hf/Nd or Hf isotopes. This fluid-modified source was then fluxed by hydrous melts, which lowered its $\delta^{98/95}Mo$, εHf, and Hf/Nd and raised its Ce/Mo (Fig. 5). The geochemistry of Pagan lavas points to a source contributor that records both fluid and melt inputs from the subducted slab. Serpentinite dragged down from the fore-arc mantle is the most likely candidate.

Figure 7 shows our preferred, Mo isotope-based model for the slab dehydration process beneath the Mariana arc. Shallow (<80 km) slab dehydration leads to partial serpentinization of the forearc mantle near the slab–mantle interface. This partly serpentinized forearc mantle is transported downward along the subducting slab, where it is further metasomatized by fluid and melt from the slab before serpentine totally breakdown at depth. Fluid/melt from the slab with high $\delta^{98/95}Mo$[32] and largely MORB like Pb–Sr isotopes[8,10] are generated by the breakdown of serpentinite in the subducted lithospheric mantle. Dehydration of

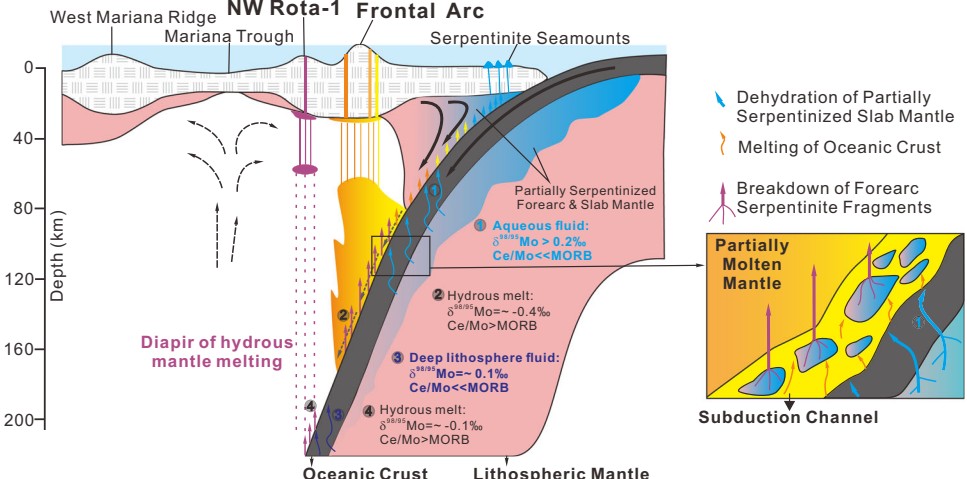

**Fig. 7 Schematic cross-section through the Mariana arc, showing slab dehydration/melting processes.** Shallow (<80 km) slab dehydration results in partial serpentinization of the forearc mantle. The partially serpentinized forearc mantle is then dragged down by the subducting slab and is further fluxed by fluid/melt from the slab at depth >80 km. Dehydration and melting of the slab were triggered by the breakdown of serpentinite in the subducted lithospheric mantle. Breakdown of multiple episodes of fluid/melt fluxed serpentinite in the subduction channel at >80 km depth released its re-homogenized fluid/melt to the overlying mantle wedge and stimulated partial melting. Serpentinite being dragged down from the mantle wedge gradually decomposed in the subduction channel and is likely became totally exhausted by 200 km depth, then a breakdown of serpentinite in the slab lithosphere control the generation of slab melt at depth >200 km.

slab lithosphere serpentinites is expected to start at low temperature as a result of brucite breakdown, e.g., 300–400 °C at pressure <2 GPa, according to results from the Erro–Tobbio meta-serpentinites in the Ligurian Alps of Italy[23,24]. This result is consistent with the estimated subducted slab Moho temperature of ~300 °C at about 100 km depth for the southern Mariana arc[56]. It is further supported by evidence from the Cabo Ortegal Complex eclogites, which have experienced long-term interaction with slab serpentinite fluids before reaching pressures >1.7 GPa and temperatures of 650–670 °C[32]. Dehydration of the slab lithosphere becomes less likely when it is not hot enough and brucite is exhausted at <165 km depth. Breakdown of serpentinites that experienced multiple episodes of fluid/melt exchange in the subduction channel at >80 km depth will release a re-homogenized fluid/melt into the overlying mantle wedge that can trigger partial melting. Melt segregation at ~75 km depth[7] generates volcanic front basalts. As partially molten mantle is only likely to homogenize on scales of tens to a few thousand meters, hydrous melt contributions are likely to increase gradually in the mantle partial melting zone with slab depth and consequently with distance from the arc magmatic front. Down dragged subduction channel serpentinites are likely to be exhausted by 200 km depth, as seen in the disappearance of the forearc serpentinite signal in NW Rota-1 lavas. At depth >200 km, the temperature of the subducted slab Moho can reach >550 °C[56] which permits antigorite to break down in the slab lithosphere[59]. The mantle wedge is thus fluxed by a slab melt with high Ce/Mo due to early Mo depletion, triggering mantle melting and the diapiric rise of hydrous melt. These hydrous melts segregate from mantle partial melting zones at ~65–50 km[36] to generate basalts behind the arc volcanic front. Our model can also account for the Mo isotope variation of other Mariana and Izu volcanic front lavas (Fig. 4). This study suggests that serpentinite dragged down from the fore-arc mantle may play an important role in material transfer from the subducted plates towards the mantle wedge. It acts as an important intermediate carrier for slab fluid and melt transfers. Future new geochemical tracers and detailed geophysical observations can test the model presented in this paper.

## Methods

**Sample selection and preparation**. The earliest record of magmatism in the Mariana subduction system is ~52 Ma from the initial convergence of the Pacific Plate toward the Philippine Sea Plate[54,60,61]. The arc later matured at ~44 Ma[62,63] and experienced rifting and back extension at ~30–15 Ma and from ~7 Ma to present (see Straub et al.[64] for review). The modern Mariana arc includes from west to east, the Mariana Trough (back-arc basin), the active Mariana arc, the old Mariana forearc (including the islands of Saipan, Tinian, Aguijan, Rota, and Guam), and the Mariana Trench (Fig. 1). The Mariana subduction system is nonaccretionary and the forearc is pervasively faulted, with numerous, large, and active serpentinite mud volcanoes derived from the shallow mantle wedge[12,13]. The active magmatic arc has been subdivided from north to south into the Northern Seamount Province, Central Island Province, and Southern Seamount Province[65]. Volcanic rock samples of this study are from Pagan and NW Rota-1 volcanos of the Mariana arc (Fig. 1). Pagan is one of the largest volcanoes along the volcanic front of the Central Island Province of the Mariana arc. It has a maximum elevation of 570 m (Mt. Pagan), with its submarine flanks descend to 2000–3000 m below sea level (Supplementary Fig. 1a). Samples from Pagan used in this study were collected in 2010 (cruise NT10-12), using ROV *Hyper-Dolphin* and R.V. *Natsushima*. Dive HPD1147 on the small knolls of NE Flank sampled two types of primitive clinopyroxene-olivine basalt (COB), COB1 (7.9–11.2 wt% MgO) and COB2 (10.1–10.9 wt% MgO), with the latter only ~500 m further away from the trench than the former. Some less evolved (8.2–9.4 wt% MgO) basalts about 1000 m SW away from the COB2 were also collected. Dive HPD1148 on the small knoll on its SW Flank recovered more evolved basalt (5.4–7.5 wt% MgO). NW Rota-1 is a submarine volcano located about 40 km west of the volcanic front of the Southern Seamount Province. It has a conical shape, with its summit at about 520 m water depth (Supplementary Fig. 1c). Its base has a diameter of about 16 km at 2700 m water depth. Samples of this study from NW Rota-1 were collected in 2005 (cruise NT05-17) and 2009 (cruise NT09-02), using ROV *Hyper-Dolphin* and R.V. *Natsushima*. Dives HPD480 and HPD481 sampled porphyritic basalt and andesite (2.8–8.2 wt% MgO) from the Summit areas at 520–1000 m depth. Dive HPD488 sampled COB (7.7–9.0 wt% MgO) and plagioclase-olivine basalt (POB; 7.8–9.4 wt% MgO) from the Eastern Flank of the volcano at 1500–2300 m depth. Dive HPD951 collected andesite lava flows (3.4–3.5 wt % MgO) from a knoll about 13 km east of the NW Rota-1 summit at 2000–2300 m depth. The Asùt Tesoru mud volcano samples examined in this study were recovered during International Ocean Discovery Program (IODP) Expedition 366. These samples were associated with upwelling pore fluids at pH > 11.7, indicating that they have not interacted with seawater (pH = ~8.1), and are thus representative of materials that have interacted with shallow, slab-derived serpentinizing fluid[13]. All samples were pulverized in an agate ball mill after sawing and jaw crushing. Major trace elements, mineral chemistry, and Sr-Nd-Hf-Pb isotopes of the volcanic rock samples of this study have been discussed in detail in Tamura et al.[36] and Tamura et al.[7].

**Mo isotope measurements**. The chemical separation and mass spectrometry measurements of Mo isotopes were conducted at the State Key Laboratory of Isotope Geochemistry, Guangzhou Institute of Geochemistry, Chinese Academy of

Sciences. In brief, an appropriate mass of sample powder was weighed out to provide 120 ng of Mo. About 120 ng of $^{97}$Mo–$^{100}$Mo double spike solution was added before digestion of the samples. The sample-spike mixture was digested by using 4 mL of HF and 2 mL of HNO$_3$ in closed beakers at 140 °C overnight. Mo separation and purification were achieved using an extraction chromatographic resin of N-benzoyl-N-phenyl hydroxylamine manufactured in-house, following the protocols of Li et al.[66] and Fan et al.[67]. After the sample matrix and interference, elements were removed, Mo isotope measurement was performed on a Thermo-Fisher Scientific Neptune-Plus multi-collector inductively coupled plasma mass spectrometer utilizing double spike analysis to correct for instrumental mass bias[68,69]. Isotope measurements are made relative to a NIST SRM 3134 standard solution. The external reproducibility of the NIST SRM 3134 standard solution is at 0.06‰ (2 SD, $n = 51$) for the $\delta^{98/95}$Mo values. USGS rock reference materials AGV-2 and W-2a and IAPSO seawater standard were simultaneously processed with the samples to monitor accuracy and gave the $\delta^{98/95}$Mo value of $-0.17 \pm 0.05$‰, $-0.03 \pm 0.06$‰ and $+2.08 \pm 0.05$‰, respectively. Repeat measurement of IAPSO seawater standard over the course of 6 months yielded a reproducibility of 2.06 ± 0.09‰ (2 SD; $n = 24$). Molybdenum concentrations were calculated from spiked isotope measurements. USGS rock reference materials AGV-2 and W-2a and IAPSO seawater standard yielded a concentration of 1.7 and 0.37 ppm and 9.5 ppb, respectively. These results of Mo isotope and concentration are consistent with certified values and with the values reported by the previous studies[66,70,71]. Repeated digestion and analysis of individual rock samples of HPD1147R13 and HPD1148R19 yielded $\delta^{98/95}$Mo difference ≤0.06‰ (Supplementary Data 1). The whole procedural blank for our analyses was 0.42 ± 0.17 ng (2 SD, $n = 3$) Mo, far less than total Mo in the samples and standards.

## Data availability
The authors declare that the data generated or analyzed during this study are included in this published article and its Supplementary Information files.

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

## Acknowledgements

The Asut Tesoru seamount samples were provided by the International Ocean Discovery Program (IODP). We thank the crew of the JOIDES Resolution, the science team, and the technical support staff for IODP Expedition 366 for their efforts in sample recovery and shipboard characterization. H.Y.L., R.P.Z., J.L., and Y.G.X. were supported by the National Natural Science Foundation of China (NSFC Project 41922020), the Strategic Priority Research Program of the Chinese Academy of Sciences (Grant No. XDB42020201, XDB18000000), and the Key Special Project for Introduced Talents Team of Southern Marine Science and Engineering Guangdong Laboratory (Guangzhou) (GML2019ZD0202). J.R. was supported by a University of South Florida UNI-Nexus award for USF-GIGCAS faculty and student exchanges, as well as via IODP post-cruise support. This is contribution No. IS–3067 to GIG-CAS and UTD Geosciences contribution No. 1674.

## Author contributions

H.Y.L. proposed the research plan, comprehensively interpreted the geochemical data, and drafted the primary paper. Y.T. supplied volcanic rock samples and the fundamental data. J.G.R. provided the serpentinite mud volcano samples and associated compositional data. R.P.Z. and J.L. conducted the Mo isotopic analyses. C.S., R.J.S., J.G.R., and Y.G.X. contribute to improving the interpretation of the data and the writing of the paper.

## Competing interests

The authors declare no competing interests.
