## [Peer Review File · Nature Communications]

Molybdenum isotopes unmask slab dehydration and melting beneath the Mariana arcReviewers' Comments:

Reviewer #1:

Remarks to the Author:

Li et al. present Mo isotopes of primitive submarine lavas from two active volcanoes in the Mariana arc to explore the issues of subducted slab devolatilization and materials transfer into the mantle wedge. These new Mo isotope data reveal that the source material of the volcanic front is first metasomatized by aqueous fluids and then fluxed by hydrous melts, whereas volcanism 40 km behind the volcanic front show the signature of a single hydrous melt slab component. They then proposed that serpentinite dragged down from the fore-arc mantle acts as an important intermediate carrier for slab fluid and melt transfer.

While I find the new Mo isotope data of high quality and the proposed model very interesting, I also think the discussion should be more thorough, some of which need quantitative calculations.

General comments:

1. In the introduction section, the authors could more stress on what is newly constrained here or refined by Mo isotope results.
2. It is better to re-assess the effects of fractional crystallization on Mo isotope and Mo/Ce ratios and re-consider the evolved samples in this study.
3. Given the authors proposed/identified several slab components based Mo isotopes and other geochemical indicators, it is useful to quantify each contribution in more detail for the samples from two volcanoes.

Overall, I think this will be a good and useful paper in the field of subduction science and Mo isotope research after revision.

More detailed comments:

Line 38 to 68: In the introduction part, the authors could more stress on which mechanism regarding this process is newly constrained here or refined by Mo isotope results.

Line 69 to 70: It is better to add refs (e.g., Bali et al., 2012, EPSL).

Line 76 to 77: Previous studies have showed that marine sediments have wide range in Mo isotopes. If the authors want to emphasize the weighted average Mo isotope composition of the sediments outboard of the Mariana arc ($\delta^{98/95}\text{Mo} = -0.29\text{‰}$; Freymuth et al., 2015), please clarify here.

Line 87 to 88: Again, it is better to add refs here.

Line 105 to 108: It is not clear why amp fractional crystallization can affect the Mo/Ce ratios of the evolved samples. This is not reported by previous studies (Voegelin et al., 2014; Wille et al., 2018). In addition, while the evolved Summit samples (yellow circle) from NW Rota-1 show similar range of MgO (2 wt% to 7 wt%) and trend of Dy/Yb to those of Pagan, they exhibit absence of correlation between Mo/Ce ratio, Mo isotopes and MgO. Furthermore, it has been suggested that fractional crystallization does not affect the Mo isotope ratios of evolved samples in other Mariana lavas with MgO ranging from 2.7 wt % to 6.1 wt % (Freymuth et al., 2015) and those from the Izu volcanic front with MgO ranging from 3.4 wt % to 5.6 wt % (Line 119 to 121). Thus, the effects of amp fractional crystallization on both Mo isotope and Mo/Ce ratios need further consideration.

I agree with the authors to concentrate on primitive submarine lavas (>7 wt % MgO) in this study to avoid the complexity introduced by fractional crystallization and/or mixing. However, on closer inspection, I find $\delta^{98/95}\text{Mo}$ of the evolved samples from both Pagan and Rota correlated with Ce/Mo and Mo/Nb. This may offer more useful information. I suggest the authors to re-consider these samples.

Line 140 to 141: It is better to keep consistency to use Ce/Mo or Mo/Ce in the whole text.

Line 192 to 193: It will be helpful to quantify each contribution in terms of Sr-Pb-Mo isotopes using simple calculation of mixing models.

Line 209: The information in Figure S1c is quite interesting and important, which needs to be included

in the main text to make it clearer. In addition, samples of SW flank from Pagan are not included in Figure S1c.

Line 222: less incompatible rather than less mobile.

Line 224 to 226: This may be problematic. From the trend of the samples from Pagan, melt derived from slab sediment is likely characterized by low $\delta^{98/95}\text{Mo}$ (-0.3 ‰) and Mo/Nb, Ba/Nb etc.

However, the trend of samples from NW Rota-1 is contrasting, which is unlikely explained by different proportion of sediment melt contribution.

Line 229: typo of "in".

Line 243: It is not clear what is the Mo isotope composition of the melt derived from the overlying subducted oceanic crust and sediments.

Figure 6: Please explain the model in detail in the caption.

Reviewer #2:

Remarks to the Author:

Heye Freymuth, review of the manuscript: Molybdenum isotopes unmask slab dehydration and melting at Mariana arc by Li et al.

The presented data build on previously published Mo isotope data for the Mariana arc and other arcs. There are several interesting aspects of the dataset that represent a step forward compared to previous publications: 1) Two individual sites are studied in detail, 2) some of the samples are from a rear-arc locality and 3) primitive samples with high MgO are included. I therefore believe that this is a valuable contribution that should be considered for publication in Nature Communications. I nevertheless found a number of issues that should be addressed before publication, in particular regarding the inference that the Mo isotope data can be used to trace serpentinized forearc mantle dragged down by the subducted slab.

The interpretation that serpentinized forearc mantle that is dragged down by the subducting slab and dehydrates to produce the geochemical fluid signature is based on the discussion lines 245-246:

".....this mechanism fails to explain the Mo isotope variations in the Pagan primitive lavas, in that neither an aqueous fluid nor a hydrous melt component shows a mixing trend with the depleted mantle component" and lines 246-255:

" Mo isotope variations and their correlation with elemental proxies for aqueous fluid and hydrous melt require that the source material for Pagan volcano was first metasomatized by an aqueous fluid phase to elevate its $\delta^{98/95}\text{Mo}$ and to decrease its Ce/Mo while keeping its Hf/Nd and Hf isotopes intact. The fluid modified source material was further fluxed by hydrous melt to decrease its $\delta^{98/95}\text{Mo}$, ϵ_{Hf} , and Hf/Nd and increase its Ce/Mo (Figures 4c and 4d). The geochemistry of Pagan primitive lavas thus need an intermediate carrier that retains both fluid and melt from the subducted slab before being released to the magma source. Serpentinite dragged down from the fore-arc mantle is the most likely candidate for this carrier."

First, Mo is fluid-mobile and highly enriched in arc magmas compared to the mantle. Thus, there is no need for mixing trends with the depleted mantle component. The fluid and slab melt components will dominate the amount of Mo in arc magmas whereas the mantle component is likely close to negligible. It should therefore be unlikely to see mixing towards the mantle.

Second, there is no need for a temporal progression in the addition of the various components (as depicted by arrows in Fig. 4). The simplest interpretation of the Pagan data and trends between $\delta^{98}\text{Mo}$ and Hf/Nd would be that the slab component added to the NE flank is more aqueous while the slab component added to the SW flank is more dominated by hydrous melt. Both are then added to the mantle, hence the more mantle-like Hf/Nd at the NE flank (as the fluid adds little Hf and Nd) and the more hydrous melt-like Hf/Nd in the SW flank (influenced by some slab melts). With such a model there is no need for interim storage of fluids in wedge serpentinites. I don't want to argue against the model of serpentinite dragged down from the fore-arc. It may well be viable. But I'm not convinced that the Mo isotope data presented here are tracing this process.

It is unfortunate that observations made in the manuscript are entirely qualitative. Mass balance and/or quantitative mixing models would be very useful here in demonstrating whether the above scenarios are viable. See e.g. models in the recent study by Villalobos-Orchard et al. for the Izu section of the IBM arc.

All Mo isotope data reported here for samples with < 7 wt. % MgO were not used later on due to concerns about effects of fractional crystallisation. Yet, published data used as reference were not filtered in the same way and in fact, previously published data are almost entirely for samples with < 7 wt. % MgO. Those studies have argued against significant modification of Mo isotope ratios by fractional crystallisation, at least for basalts and basaltic andesites. In the Pagan data presented here, only two samples are significantly isotopically lighter than the rest and interestingly, a similar "trend" does not exist for NW Rota. While it is clearly best to focus on primitive samples (and this study reports some of the very few Mo isotope data for high MgO samples) it seems arbitrary to selectively ignore some of the samples with lower MgO.

There is no information on sample preparation in the method section. This is particularly important because the samples are from submarine eruptions and hence easily altered by seawater.

Fig. 3A shows a much steeper trend in $d_{98}\text{Mo}$ vs. Ce/Mo for Pagan than for previously published arc sections, suggesting an isotopically heavier fluid. This is an interesting aspect of the data, in particular with regards to the mass balance of Mo in subduction zones and beyond and thus worth highlighting.

Other comments on lines:

46-47: It's odd to introduce the fluids as derived from altered oceanic basalt and then later discussing them as derived from serpentinites. That classic AOC model is not really up to date any more.

136-137: Only one Pagan sample has higher Ce/Mo than the depleted mantle in Fig. 3a.

146: Radiogenic Hf and Nd in the mantle doesn't need metasomatic input.

179-180: Why just sediment melt? The AOC may well melt, too.

188: "To reconcile this dilemma..." I think this dilemma needs to be explained in more detail and assessed quantitatively, in particular with respect to the subsequent sentence stating that the sediment contribution is "minimal".

196-198: This agrees well with a similar model we have proposed for the Izu section of the arc (Freymuth et al. 2016 GCA 186 and Freymuth et al. 2019 EPSL 522).

Reviewer #3:

Remarks to the Author:

Review of MS NCOMMS-21-02686 "Molybdenum isotopes unmask slab dehydration and melting at Mariana arc"

Reviewed by Ivan Savov (Univ. Leeds)

This is a very well written manuscript, with excellent graphics and good data quality, including data of a novel tracer (Mo isotopes). The manuscript supports interesting and currently debated hypothesis for the mechanisms of elemental and isotope cycling from subducting plates to the surface volcanoes.

The insights from the manuscript can be largely applicable to multiple and diverse disciplines of the earth and marine sciences. With that said, I think with moderate revisions the manuscript will have high and overall positive impact and is worth publishing in a journal such as Nature Communications.

Main comments:

The manuscript contains both new Mo isotope data, as well as published isotope and elemental datasets on these same rock samples. Some of the authors are world-leading experts on arc geochemistry and not surprisingly their sample selection is excellent. The differences with the previous Mo isotope datasets from the same arc (*but different volcanoes) are a bit worrisome, but the explanation of the authors is quite convincing and not analytical in nature.

Some of the conclusions and ideas in this manuscript are in direct clash with the views of Chen et al, Nature Comm. 2019 (you have this reference on your line 390), which is also involving Mo isotopes and review of the role of serpentinites (from an Raspas, which is exhumed and well preserved ophiolite complex). In a way this is great and good for your story, making the interpretation novel. In fact your story is basically like their figure 4, but turned upside down. Also worth noting is their AOC compositions as shown on a Mo isotopes vs Mo/Ce plot, where AOC seems to be off the trends shown by the ophiolite, which is in itself not supporting the altered crust/AOC as a "player". I particularly agree that there are some really important insights in the correlations seen and their careful consideration does lead to the conclusions that the hydrated in the forearc and previously depleted mantle peridotites are an end member in the arc magma sources. This has been suggested more than a decade ago via trace element arguments, with Pb-Nd-B isotopes (Tera and co-workers, Ishikawa-san & Nakamura-san; Ryan and co-workers, among others). Adding another tracer for support of the forearc mantle contribution to arc magmas is outstanding achievement! It is novel in that the data is extending the variations of arc volcanic rocks previously reported and via combination of FME/Nb ratios it is convincingly supporting a widely debated issue of the type of serpentine input into the arc magmatic source- lithospheric in origin (deep MORB mantle at bottom of gabbroic slabs) or forearc modified and down dragged with the slab to depths and with ultimately (ultra)-depleted mantle protoliths. I urge the authors to browse through the recently published paper (in Nature Comm.) that reports on the modelling of fluid penetration in deep slabs to form serpentinites and the d11B signatures of the altered oceanic crust (that is eventually subducted). This study (reference is below) is another independent evidence for lack of arc contributions from the hydrated lithospheric section of the slabs. This fact is leaving the forearc fluid-modified mantle as the only other viable alternative (and more reasonable to be honest as hydrous slabs will be hard to subduct due to low density!). McCaig, A.M., et al. No significant boron in the hydrated mantle of most subducting slabs. Nature Comm. 9, 4602 (2018). <https://doi.org/10.1038/s41467-018-07064-6>

The conclusions of (this) manuscript are also similar to the study of Kimura-san, where there is fantastic and very quantitative estimates of the nature of inputs from the slab and mantle under cross arc volcanic chains across the Izu arc (fluid X= 2-4% shallowly and meltX=1.5-4.5% for the deep melts). Their conclusions were derived from trace elements and also from Sr and Pb isotopes in combination with trace elements. All such data is apparently available for the mafic rocks from the Marianas (this study), so some links perhaps can be further established. I accent on the idea of the manuscript to be a bit more quantitative. I suggest that the latest version of arc basalt simulator (ABS) can be used to show some parallels with the Kimura-san's 2010 study (doi:10.1029/2010GC003050) . Perhaps there may be some bridges that can prove useful and importantly some quantifications from the Marianas may be revealing a more global case.

Your figures 3-bottom two segments with Ba/Nb and Cs/Nb- here one may argue that there are two trends, but those trends are not necessarily the one you highlight. If we want to include the Izu arc in the discussion, it will be easier to have trend 1[steep trend] consisting of all NWRota◇Pagan (COB 2)◇the high Ba/Nb samples from Alamagan and Guguan◇all of the Izu arc data (with a mixing end member DM); and trend 2 (more vertical one) = including Agrigan and Uracas volcanoes◇NW Pagan and the highest 98/95 Mo samples from Alamagan and Guguan volcanoes.

I urge the authors to take the forearc serpentinitized mantle trace elements from, say Savov et al., 2007-JGR and see where on the trace element graphs these potential end-member compositions

will/may plot. Site 801C is clearly not very useful to explain the mixing relationships and so adding something in the high Ba/Nb and Cs/Nb end of the plot and always with very high 98/95 Mo will be quite useful. What is in the upper right corner of your diagrams, anyway? Why is Izu arc and the NE flank of Pagan trending in this direction? Please also note that there may be two different trends on your figure 4 (Mo isotopes vs. Hf isotopes). What is sitting in the upper right corner there? Please add an end-member on the plot. Again the trends here do not go through 801C basalt composite. I suppose that some of the highly modified serpentinites will have elevated 98/95Mo (aqueous fluid arrows on the other plots show vertical enrichments of 98/95Mo) and so these may be good to show (or if anyone have looked at serpentinites for Mo isotopes- those will be good to see as they are central to your hypothesis).

There is now published dataset for adakitic melts, effects of hornblende in the source of melts and Mo isotopes. The work is published in *Geochim Cosmochim. Acta* (<https://doi.org/10.1016/j.gca.2021.01.020>). It appears to show that hornblende crystallization may dominate the Mo isotope variations in arc magmas. Please consider this new dataset, especially when you discuss the deep sourced magmas. There may be useful information about the effect of mineralogy and mantle metasomatism. Also please check if there is anything in the literature on Mo isotopes in metasomatized/serpentinized rocks.

There are small details that I think may be helpful to clarity, especially for the non-expert (petrologists and arc geochemist) audience of *Nature Comm*. I list below some of those points and some of the text edits that will help:

Minor details linked to the text.

line 39- will help if you state "intraoceanic". The mentioning of end-membr convergent margin is not helpful. I rather add that it is "non accreting" and also add that it is quite sediment starved, making it appropriate, if not unique for the cross arc cycling studies.

line 47- no CAPS for Boron

line 48- The cited studies does not involve Hf isotopes. Perhaps the study of J.Pearce on the Hf isotopes of the Marianas will be good to add.

line 50- Please note that the Nd and Hf are NOT a good tracers of fluids as the elements in question are highly fluid immobile. In the case of Nd- please note that there is increasing amount of evidence in the literature that the 143/144Nd of serpentinites may indeed vary vastly (research by Bizimis and co-workers) and the process behind it is not well understood.

line 53- "inconsistent with the Mariana arc lavas"- perhaps add the range here.

Line 54- these are indeed some moderately elevated 87/86Sr values, but this is in respect to MORBs. Otherwise, in respect to anything in the slab those are immensely low or better- unradiogenic.

line 74-77- what are the errors for the Mo isotope ratios. If those are large, then the MORBs and sediments may nearly overlap. This is critical point why we need AOC as end-member and why you have shown exactly that on your plots (ODP Site 801 end member).

Line 84- You need to state how we know these depths. What methods were used to determine where the slabs are and at what dip they sink in the mantle. A reference will also be good.

line 108- East Knoll is with CAPS.

line 112- you may want to state somewhere what is the slab DIP. One way to do this is maybe in the caption of your schematic summary diagram. In any case- for non specialists there is a need to explain that. Another option will be to add it as a method.

line 125- but this is not too distant range I respect to the Izu VF (0.16 per mil) , isn't it? This is one of the reasons that the range of the isotope errors needs to be properly reported.

Line 132- Cs/Nb are high in fluid, no doubt. This has been shown nicely in Savov et al, 2007-JGR manuscript.

Line 140- I recommend that for the same of consistency you stick to your Ce/Mo (as in your line 126) and do not confuse things by introducing Mo/Ce. So here you may say " high Ce/Mo".

line 168- elevated Sr isotope ratios in respect to MORB. And only so very little!

line 171-172- Need to tell us if this fluid is realistic (see my recommendation for use of McCaig et al,

2019) . In any case- if this fluid exists [doubt that!] is released- it will react with the abundant fresh gabbro and diabase and will not manage to do anything. If you eliminate this option early- then you can use some lines for properly introducing the forearc down dragging processes (which are not vague) and link to the Mariana md volcanoes, which are physical evidence for the high Ba and Ca you need.

Line 180- why is this. Maybe clarify your arguments.

line 184- this increasingly make more and more sense to me.

line 193 (and several other places in the manuscript)- a just published GCA paper by Anders et al (2021) is reporting very interesting story from Izu arc and is telling us tat sediments are not playing a role and that the slabs are melting. Please add this paper insights (Sr, Pb, Nd, Hf isotopes are VERY abundant!) to your story. I think it is highly relevant to see what is the entire range of mantle melts, nicely shown vs. sediments- all nicely summarized in their plots.

line 196- need reference here after t"sediments"

line 229- "from in"

line 232- hence the two trends on figure 4B (Hf isotopes)

line 236- This is a good place to cite some forearc serpentinite peridotite major and trace element paper.

line 239- it will be more thorough if here you also cite McCaig et al. (2019).

Line 246- here refer to some of your figures

line 254-this "dragging down" of the serpentinized mantle needs to be either explained in a bit more detail or some key references need to be given. As it is, for average reader, this is not clear enough.

line 255- this is SUPER! I really think this is a great selling point of your paper and you may want to further accent on this fact, in addition to the mixing trends and arc magma genesis. People should start plotting this as mixing end member and not some composite samples of AOC or sediments, which may but may not be relevant.

line 263- "Breakdown of serpentine ..."- see McCaig et al. (2019). Also note than at high T this breakdown will lead to formation of chlorite- rich protoliths.

Line 272- please give reference here for the depth.

Just a note to the authors, that serpentinites often have Ba/Th (10^3 and 10^4).

DETAILED RESPONSES TO THE REVIEWERS

Overall response to reviewers:

The reviewer's main comments are answered as follows:

(1) Key contribution.

Our key contribution in this study is that we can use the Mo isotopes to discriminate among different sources of subduction related serpentinites. Forearc mantle and slab lithosphere serpentinites play different roles in generating subduction zone fluids. On the one hand, forearc serpentinites downdragged by the slab act as an important intermediate carrier for slab fluid and melt transfer in the subduction channel. On the other hand, dehydration of serpentinites in the slab lithosphere is critical for triggering slab dehydration and melting at depth. We have rewritten part of the introduction and discussion sections. The following information has been added in the revision:

Recent geochemical and geophysical observations proposed that dehydration of serpentinite may play a more critical role in explaining these observed elemental and isotopic paradoxes (Cai et al., 2018; Savov et al., 2005, 2007; Tamura et al., 2014; Freymouth et al., 2015). Two kinds of serpentinite may have contributed fluids for the Mariana arc mantle. The first is serpentinite formed in the shallow forearc mantle wedge as a result of the incorporation of low-temperature fluid (80-350°C) from the slab during its initial burial (<19km; Freyer et al., 1996, 2020; Savov et al., 2005, 2007). Forearc mantle serpentinite that diapirs and extrudes at the seafloor are extremely enriched of fluid mobile elements (e.g., B, As, Cs, Sb, Li) and has very high $\delta^{11}\text{B}$ (> 15‰), indicating large slab inventory depletion of B (>75%), Cs (>25%),

As (>15%), Li (>15%), and Sb (>8%) during its initial stage of dehydration and significant residual slab $\delta^{11}\text{B}$ ($\sim 6\pm 4\%$) decreasing as a result of selective removal of heavy ^{11}B by the fluid (Benton et al., 2001; Savov et al., 2005, 2007; Pabst et al., 2012). The slab would experience further fluid mobile elements depletion and $\delta^{11}\text{B}$ decreasing during deeper subduction. Physically addition of forearc serpentinites to the sub-arc mantle or subduction channel would be required to explain the enrichment of fluid mobile element and heavy B isotope (e.g. +4.5 to +12.0‰ in the Izu Arc; Ishikawa and Nakamura, 1994; Straub and Layne, 2002) characteristics of the volcanic front basalt (Benton et al., 2001; Savov et al., 2005, 2007; Tonarini et al., 2011; Ryan and Chauvel, 2014).

The second kind of serpentinite is that formed by seawater infiltrating oceanic mantle lithosphere via normal faults in response to plate bending as it enters the trench (Cai et al., 2018). Although the chemical composition of serpentinites in the slab mantle is not well understood, it depends on how seawater enters the lithospheric mantle and how the lithosphere filters elements from the fluid (McCaig et al., 2018), hydrous minerals, e.g. brucite and antigorite, in the serpentinite would breakdown when they are heated up to 300-400 °C (Plümper et al., 2016; Peters et al., 2020). The channelized serpentinite fluid may be heated up when migrating up to the slab surface and incubate the ability to extract the Pb-Sr-Mo elements from the crust (like Pb, Ba, and Sr), but can only have a very low concentration of fluid immobile elements (like Nd and Hf; Kessel et al., 2005). It is a dynamic equilibrating process for the fluid traveling through the overlying whole section of the oceanic crust before

it finally triggers melting of the slab surface or being directly delivered into the magma source. Therefore, the deep slab sourced serpentinite fluid would potentially modulate the total slab fluid/melt Sr-Pb isotopes toward the pristine MORB signature and generate a fluid mobile elements enriched signature (Tamura et al., 2014; Freymouth et al., 2015, 2016, 2019; Klaver et al., 2020).

Whilst the importance of serpentinite subduction in the subduction zone chemical cycle is now widely acknowledged, the details of which kind of serpentinites are responsible and how they work in subduction zones are still unclear (Spandler and Pirard, 2013; McCaig et al., 2018). Our research reveals that both kinds of serpentinite stated above are important. On the one hand, forearc serpentinite being dragged down by the slab may act as an important intermediate carrier for slab fluid and melt transfer in the subduction channel. On the other hand, dehydration of serpentinite in the slab lithosphere is critical for triggering slab dehydration and melting at a deep depth.

(2) Quantitative calculation.

A major criticism from the referees is the qualitative character of our original explanation. In the revision, we did detailed quantitative calculations using the Mo-Nd-Hf elemental and isotopic compositions for slab components (sediments, altered MORB, fresh MORB, and MORB-like eclogite) and the ambient mantle, partition coefficients during slab dehydration/melting at different P-T condition (Kessel et al., 2005; Adam and Green, 2006; Bali et al., 2012; Chen et al., 2019), and

Mo isotope fractionation factors during slab dehydration/melting (Chen et al., 2019). The Mo-Nd-Hf mobility in the slab fluids at different temperatures is fully considered in these calculations. Our quantitative calculations support and clarify our original interpretation. Calculation details are as follows:

(a) Based on Hf-Nd isotopes and Hf/Nd ratios of the arc basalt and the ambient mantle (Woodhead et al., 2012 *Geology*), we first estimate that the melts component of the slab contains both contributions of sediments and altered mafic oceanic crust (AMOC) with a ratio of 1: 9 (Figs. 5c and S3).

(b) The shallow slab fluid (700°C; Source = 90% AMOC + 10% Sediments; F=2%) composition is calculated generally according to the methods in Villalobos-Orchard et al. (2020), with Mo content being calculated according to the uniform Ba/Mo=230 for the Pagan Northeastern Flank samples and the calculated Ba content for the fluid.

(c) The Ce/Mo and Mo isotopes of the average eclogite from Chen et al. (2019) were used to represent the bulk subducted slab that experienced long-term interaction with deep serpentinite fluid (Chen et al., 2019). Therefore, the Mo of the subducted slab is scaled to Ce/Mo=80 according to the average eclogite from Chen et al. (2019). Based on the available parameters, our calculations reveal melting of the subducted slab ($\delta^{98/95}\text{Mo}=-0.45\text{‰}$; average eclogite from Chen et al. 2019) at 900°C can generate a melt with $\delta^{98/95}\text{Mo}$ of -0.35‰, similar to the lowest $\delta^{98/95}\text{Mo}$ of the Pagan samples (-0.31‰). Our calculations reveal the Pagan volcano mantle source contains ~0-6‰ shallow aqueous fluids ($\delta^{98/95}\text{Mo}$ of ~ 0.25‰; Villalobos-Orchard et al., 2020)

and 1-9 % hydrous melts from the subducted slab. Our calculations reveal the shallow aqueous fluids contribution decreased while the slab melt component increased along with slab depth increasing for the volcanic front samples.

(d) The $\delta^{98/95}\text{Mo}$ versus Hf/Nd correlation gives important constraints on how these slab components, i.e., shallow fluid and slab melt, were added to the Pagan magma source materials. The best and most plausible explanation is that the source material of the Pagan samples is first metasomatized by aqueous fluids with high $\delta^{98/95}\text{Mo}$, and is then fluxed by hydrous melts with low $\delta^{98/95}\text{Mo}$, Hf/Nd, ϵNd , and ϵHf . The aqueous fluids signature is best explained by dragging down of forearc serpentinitized mantle by the slab. The idea of 'no need for interim storage of fluids in wedge serpentinites' is essential that Mo-Nd-Hf variations of the basalts are only controlled by the corresponding slab surface temperature, i.e., slab input with a more fluid signature gradually evolved to a more hydrous melt signature along with increasing of slab depth. The NE Pagan samples have very limited corresponding slab depth variation, 124.5km to 129 km (Source data; Calculated according to the Slab2 data of Hayes et al. 2018). It is difficult to imagine the slab input evolve significantly in such a narrow range of slab depth. Our calculations reveal a sole low-temperature slab fluid input is difficult to explain the low Hf/Nd and high $\delta^{98/95}\text{Mo}$ characteristics of the Pagan Northeastern Flank samples. It is revealed that their mantle source would need ~30-40% input of a slab fluid dehydrated at 700°C or ~20% input of a slab fluid dehydrated at 800°C to decrease their Hf/Nd from a value similar as the depleted mantle as there is only very low Hf and Nd contents in the 700-800°C slab fluid. This

kind of high percentage of slab input is impossible to happen in nature.

(e) For the NW Rota-1 samples except those from the East Knoll, the $\delta^{98/95}\text{Mo}$ versus Hf/Nd correlation indicates their mantle source is metasomatized by a single hydrous melt slab component. Our calculations reveal the average eclogite mixed with 2% fluid ($\delta^{98/95}\text{Mo}=0.29\text{‰}$) that traveled through and equilibrated with the deep crust with a fresh MORB like Mo isotope composition (original $\delta^{98/95}\text{Mo}=-0.21\text{‰}$) at 700°C can generate a new slab component with $\delta^{98/95}\text{Mo}=-0.1\text{‰}$. Melting of this new slab component can generate a slab melt with $\delta^{98/95}\text{Mo} = \sim -0.01 \text{‰}$. This is good evidence for serpentinite breaking down to generate fluid that triggers melting of the slab surface for the deeply subducted slab. This explanation for the Mo isotopes also meets the low $^{87}\text{Sr}/^{86}\text{Sr}$ signature of the NW Rota-1 samples. The low Ce/Mo and high Hf/Nd signatures of the East Knoll samples indicate their mantle source contains mostly hydrous fluid if these samples haven't experienced significant fractional crystallization of amphibole (Figs 4e and 5d).

(3) The effects of fractional crystallization on Mo isotope and Mo/Ce ratios.

The basic law for fractional crystallization of amphibole affecting the Mo/Ce of the magma is that Mo is more incompatible than Ce for a given amphibole-bearing magma system. It is confirmed by the trace element partitioning during amphibole-bearing garnet lherzolite melting (Adam and Green, 2006) and magma differentiation of the Kos Plateau Tuff (Voegelin et al., 2014). Also, amphibole preferentially incorporates middle REEs over heavy REEs, as well as Nd over Hf

during oceanic crust melting or magma fractional crystallization (e.g., Tiepolo et al., 2007). Therefore, amphibole fractional crystallization will induce increasing of Hf/Nd and decreasing in Ce/Mo and Dy/Yb of the residual magma.

For our Southwestern Flank samples of Pagan, we can see the amphibole fractional crystallization trend based on Hf/Nd, Ce/Mo, and Dy/Yb versus MgO (Or SiO₂) correlations. Samples with MgO < 7% have been excluded in the following diagrams (Figs 3-6).

For the Summit and Eastern Flank samples of the NW Rota-1, no obvious decrease of Ce/Mo and Dy/Yb and increase of Hf/Nd can be observed with decreasing of MgO, therefore amphibole fractional crystallization is not significant for these samples. All these samples have been included in the following diagrams (Figs 3-6). It is difficult to assess if samples from the East Knoll of the NW Rota-1 have experienced significant amphibole fractional crystallization, as they have very limited MgO variation (Figure 2). They are also discussed in the following text and diagrams. The low Ce/Mo and high Hf/Nd signatures of the East Knoll samples indicate their mantle source contains mostly hydrous fluid if these samples haven't experienced significant fractional crystallization of amphibole (Figs 4e and 5d).

Including the eight additional samples from the NW Rota-1 in the revision doesn't affect the original interpretations.

Amphibole preferentially holds onto the lighter Mo isotopes (low $\delta^{98/95}\text{Mo}$)

during fractional crystallization (Voegelin et al., 2014; Wille et al., 2018). Thus, amphibole fractional crystallization should induce a negative correlation between Ce/Mo and $\delta^{98/95}\text{Mo}$ for the residual magma. This correlation is like the Ce/Mo versus $\delta^{98/95}\text{Mo}$ correlation induced by slab dehydration (Freymuth et al., 2015; Villalobos-Orchard et al., 2020). Therefore, samples that experienced significant amphibole fractional crystallization should be excluded before discussing the slab dehydration and mantle melting process.

For other Izu and Mariana data cited in the text and diagrams, we didn't find the amphibole fractional crystallization effects based on proxies such as Hf/Nd and Dy/Yb. We think these samples didn't experience significant fractional crystallization of amphibole, thus their Ce/Mo and Mo isotopes mostly reflect the slab dehydration and mantle melting processes.

(4) Mo isotopes of the forearc serpentinites.

We analyzed Mo isotopes for serpentinite mud samples from the Asùt Tesoru mud volcano (collected during IODP Exp. 366; informally known as Big Blue Seamount) during the review process, and the data are presented in the Source Data document and now plotted in Fig. 3. The Asùt Tesoru mud volcano is located about 72 km west of the trench axis, 134 km east of the Pagan volcano, and lies about 18 km above the downgoing plate (Fryer et al., 2020). The samples of this study have corresponding pore fluid with pH > 11.7 and high Mo concentrations (Wheat et al 2018), indicating they are not affected by the seawater (pH = ~8.1) after erupted above the seafloor,

thus may closely represent shallow slab fluids (Fryer et al., 2020). These serpentinite samples have high $\delta^{98/95}\text{Mo}$ from 0.19‰ to 0.52‰ and low Ce/Mo from 0.5 to 5.5, indicating the forearc mantle serpentinite could be potential source components for the IBM basalts (Fig. 3).

We should point out that the forearc serpentinites generated at shallow depth will be further metasomatized by the slab fluid when they are dragged down by the slab to a depth of > 80 km, where the slab starts to couple with the overriding mantle and being heated up to devolatilize. The mantle wedge serpentinite we discriminated is not exactly equivalent to serpentine mud volcanoes that erupt at the forearc corresponding to a very shallow plate depth (< 19 km).

This information have been added in the revision.

Reviewer #1 (Remarks to the Author):

Li et al. present Mo isotopes of primitive submarine lavas from two active volcanoes in the Mariana arc to explore the issues of subducted slab devolatilization and materials transfer into the mantle wedge. These new Mo isotope data reveal that the source material of the volcanic front is first metasomatized by aqueous fluids and then fluxed by hydrous melts, whereas volcanism 40 km behind the volcanic front show the signature of a single hydrous melt slab component. They then proposed that serpentinite dragged down from the fore-arc mantle acts as an important intermediate carrier for slab fluid and melt transfer.

While I find the new Mo isotope data of high quality and the proposed model very

interesting, I also think the discussion should be more thorough, some of which need quantitative calculations.

General comments:

1. In the introduction section, the authors could more stress on what is newly constrained here or refined by Mo isotope results.

Please see reply to the 'Overall response to reviewers (1)'.

2. It is better to re-assess the effects of fractional crystallization on Mo isotope and Mo/Ce ratios and re-consider the evolved samples in this study.

Please see reply to the 'Overall response to reviewers (3)'.

3. Given the authors proposed/identified several slab components based Mo isotopes and other geochemical indicators, it is useful to quantify each contribution in more detail for the samples from two volcanoes.

Please see reply to the 'Overall response to reviewers (2)'.

Overall, I think this will be a good and useful paper in the field of subduction science and Mo isotope research after revision.

More detailed comments:

Line 38 to 68: In the introduction part, the authors could more stress on which mechanism regarding this process is newly constrained here or refined by Mo isotope

results.

We have rewritten part of the introduction section, please see reply to **Overall response to reviewers (1)**.

Line 69 to 70: It is better to add refs (e.g., Bali et al., 2012, EPSL).

Cited. We added detailed calculation for the slab fluid based Mo partition information from Bali et al. (2012) in the revision.

Line 76 to 77: Previous studies have showed that marine sediments have wide range in Mo isotopes. If the authors want to emphasize the weighted average Mo isotope composition of the sediments outboard of the Mariana arc ($\delta^{98/95}\text{Mo} = -0.29\text{‰}$; Freymuth et al., 2015), please clarify here.

We use AMOC to represent the altered mafic oceanic crust in the revision. The ranges of AMOC and sediments $\delta^{98/95}\text{Mo}$ have been added.

Line 87 to 88: Again, it is better to add refs here.

References of Tamura et al. (2011, 2014) have been added here.

Line 105 to 108: It is not clear why amp fractional crystallization can affect the Mo/Ce ratios of the evolved samples. This is not reported by previous studies (Voegelin et al., 2014; Wille et al., 2018). In addition, while the evolved Summit samples (yellow circle) from NW Rota-1 show similar range of MgO (2 wt% to 7

wt%) and trend of Dy/Yb to those of Pagan, they exhibit absence of correlation between Mo/Ce ratio, Mo isotopes and MgO. Furthermore, it has been suggested that fractional crystallization does not affect the Mo isotope ratios of evolved samples in other Mariana lavas with MgO ranging from 2.7 wt % to 6.1 wt % (Freymuth et al., 2015) and those from the Izu volcanic front with MgO ranging from 3.4 wt % to 5.6 wt % (Line 119 to 121). Thus, the effects of amp fractional crystallization on both Mo isotope and Mo/Ce ratios need further consideration.

I agree with the authors to concentrate on primitive submarine lavas (>7 wt % MgO) in this study to avoid the complexity introduced by fractional crystallization and/or mixing. However, on closer inspection, I find $\delta^{98/95}\text{Mo}$ of the evolved samples from both Pagan and Rota correlated with Ce/Mo and Mo/Nb. This may offer more useful information. I suggest the authors to re-consider these samples.

Please see reply to the **Overall response to reviewers (3)**.

Line 140 to 141: It is better to keep consistency to use Ce/Mo or Mo/Ce in the whole text.

We select to use Ce/Mo consistently in the text and diagrams.

Line 192 to 193: It will be helpful to quantify each contribution in terms of Sr-Pb-Mo isotopes using simple calculation of mixing models.

We think detailed quantify calculation based on Sr-Pb isotope is difficult as the slab surface would experience long-term, and dynamic interaction with the serpentinite

fluid from the deep slab lithosphere (Chen et al., 2019; Freymuth et al., 2015; Klaver et al., 2020). These processes affect the Sr-Pb-Mo elements and isotopes a lot. In our calculation, we use Hf-Nd-Mo systematics to constrain the contributions from different source, as Hf-Nd is hardly to be affected by the serpentinite fluid. Please see reply to the '**Overall response to reviewers (2)**'.

Line 209: The information in Figure S1c is quite interesting and important, which needs to be included in the main text to make it clearer. In addition, samples of SW flank from Pagan are not included in Figure S1c.

This diagram has been moved to the main text (Figure 4 now).

Line 222: less incompatible rather than less mobile.

After detailed consideration, we think it is not a partition effect. The reason is the slab is already depleted of Mo when it heated up to melt after early dehydration at shallow depth. It is supported by the high Ce/Mo of the eclogite that didn't experience melting (Chen et al., 2019) according their metamorphic temperature ($\sim 600^\circ\text{C}$) and their trace elemental signatures. We have corrected the statements in the whole text.

Line 224 to 226: This may be problematic. From the trend of the samples from Pagan, melt derived from slab sediment is likely characterized by low $\delta^{98/95}\text{Mo}$ (-0.3 ‰) and Mo/Nb, Ba/Nb etc. However, the trend of samples from NW Rota-1 is contrasting, which is unlikely explained by different proportion of sediment melt contribution.

Please see reply to the ‘**Overall response to reviewers (2)**’. Based on the available parameters, our calculations reveal melting of the average eclogite ($\delta^{98/95}\text{Mo}=-0.45\%$) from Chen et al. (2019) at 900°C can generate a melt with $\delta^{98/95}\text{Mo}$ of -0.35%, similar as the lowest $\delta^{98/95}\text{Mo}$ of the Pagan samples. For the NW Rota-1 samples, the $\delta^{98/95}\text{Mo}$ versus Hf/Nd correlations indicate their mantle source is metasomatized by a single hydrous melt slab component. Clearly, melting of the average eclogite from Chen et al. (2019) cannot get a melt with $\delta^{98/95}\text{Mo} > -0.1\%$. Our calculations reveal the average eclogite mixed with 2% fluid ($\delta^{98/95}\text{Mo}=0.29\%$) that extensively equilibrated with the deep crust with a fresh MORB like Mo isotope composition (original $\delta^{98/95}\text{Mo}=-0.21\%$) at 700°C can generate a new slab component with $\delta^{98/95}\text{Mo}=-0.1\%$. Melting of this new slab component can generate a slab melt with $\delta^{98/95}\text{Mo} = \sim -0.01 \%$. This is a good evidence for serpentinite breaking down to generate fluid that flux the slab surface to melt for the deeply subducted slab. This explanation for the Mo isotopes also meets the low $^{87}\text{Sr}/^{86}\text{Sr}$ signature of the NW Rota-1 samples.

The information has been added in the text.

Line 229: typo of “in”.

Corrected.

Line 243: It is not clear what is the Mo isotope composition of the melt derived from the overlying subducted oceanic crust and sediments.

Please see reply to the '**Overall response to reviewers (2)**'. The Mo isotope information is added in the text.

Figure 6: Please explain the model in detail in the caption.

Added.

Reviewer #2 (Remarks to the Author):

Heye Freymuth, review of the manuscript: Molybdenum isotopes unmask slab dehydration and melting at Mariana arc by Li et al.

The presented data build on previously published Mo isotope data for the Mariana arc and other arcs. There are several interesting aspects of the dataset that represent a step forward compared to previous publications: 1) Two individual sites are studied in detail, 2) some of the samples are from a rear-arc locality and 3) primitive samples with high MgO are included. I therefore believe that this is a valuable contribution that should be considered for publication in Nature Communications. I nevertheless found a number of issues that should be addressed before publication, in particular regarding the inference that the Mo isotope data can be used to trace serpentinized forearc mantle dragged down by the subducted slab.

General comments:

The interpretation that serpentinized forearc mantle that is dragged down by the subducting slab and dehydrates to produce the geochemical fluid signature is based on the discussion lines 245-246: ".....this mechanism fails to explain the Mo isotope

variations in the Pagan primitive lavas, in that neither an aqueous fluid nor a hydrous melt component shows a mixing trend with the depleted mantle component” and lines 246-255: “ Mo isotope variations and their correlation with elemental proxies for aqueous fluid and hydrous melt require that the source material for Pagan volcano was first metasomatized by an aqueous fluid phase to elevate its $\delta^{98/95}\text{Mo}$ and to decrease its Ce/Mo while keeping its Hf/Nd and Hf isotopes intact. The fluid modified source material was further fluxed by hydrous melt to decrease its $\delta^{98/95}\text{Mo}$, ϵ_{Hf} , and Hf/Nd and increase its Ce/Mo (Figures 4c and 4d). The geochemistry of Pagan primitive lavas thus need an intermediate carrier that retains both fluid and melt from the subducted slab before being released to the magma source. Serpentinite dragged down from the fore-arc mantle is the most likely candidate for this carrier.”

First, Mo is fluid-mobile and highly enriched in arc magmas compared to the mantle. Thus, there is no need for mixing trends with the depleted mantle component. The fluid and slab melt components will dominate the amount of Mo in arc magmas whereas the mantle component is likely close to negligible. It should therefore be unlikely to see mixing towards the mantle.

Second, there is no need for a temporal progression in the addition of the various components (as depicted by arrows in Fig. 4). The simplest interpretation of the Pagan data and trends between $\delta^{98}\text{Mo}$ and Hf/Nd would be that the slab component added to the NE flank is more aqueous while the slab component added to the SW flank is more dominated by hydrous melt. Both are then added to the mantle, hence the more mantle-like Hf/Nd at the NE flank (as the fluid adds little Hf and Nd) and the more

hydrous melt-like Hf/Nd in the SW flank (influenced by some slab melts). With such a model there is no need for interim storage of fluids in wedge serpentinites. I don't want to argue against the model of serpentinite dragged down from the fore-arc. It may well be viable. But I'm not convinced that the Mo isotope data presented here are tracing this process.

It is unfortunate that observations made in the manuscript are entirely qualitative. Mass balance and/or quantitative mixing models would be very useful here in demonstrating whether the above scenarios are viable. See e.g. models in the recent study by Villalobos-Orchard et al. for the Izu section of the IBM arc.

We thank these great comments and good suggestions. We have added detailed calculation for the different slab components based on Mo-Nd-Hf systematics. Please see reply to the **Overall response to reviewers (2)**.

A most important finding of this study is the covariations between Hf-Nd isotopes, Hf/Nd ratios and $\delta^{98/95}\text{Mo}$. The Hf/Nd variation of the basalts is sensitive to the slab fluid temperature. Calculations based on Mo-Nd-Hf systematics have advantages that the Hf-Nd mobility are much better understood (e.g. Kessel et al., 2005) thus can give some solid constraints.

The idea of '**no need for interim storage of fluids in wedge serpentinites**' is essentially that Mo-Nd-Hf variations of the basalts is only controlled by the corresponding slab surface temperature, i.e., slab input with a more fluid signature gradually evolved to a more hydrous melt signature along with increasing of slab depth. The Pagan Northeastern Flank samples have very limited corresponding slab

depth, 124.5km to 129 km (Source data; Calculated according the Slab2 data of Hayes et al. 2018). It is difficult to imagine the slab input evolve significantly in such a narrow range of slab depth. Our calculations reveal a sole low temperature slab fluid input is difficult to explain the low Hf/Nd and high $\delta^{98/95}\text{Mo}$ characteristics of the Pagan Northeastern Flank samples. It is revealed that their mantle source would need ~30-40% input of a slab fluid dehydrated at 700°C or ~20% input of a slab fluid dehydrated at 800°C to decrease their Hf/Nd from a value similar as the depleted mantle as there is only very low Hf and Nd contents in the 700-800°C slab fluid. This kind of high percentage of slab input is impossible to happen in nature.

The best and most plausible explanation is that the source material of the Pagan samples is first metasomatized by aqueous fluids with high $\delta^{98/95}\text{Mo}$, and is then fluxed by hydrous melts with low $\delta^{98/95}\text{Mo}$, Hf/Nd and ϵHf . The aqueous fluids signature is most possibly inherited from the forearc serpentized mantle dragged down by the slab.

All Mo isotope data reported here for samples with < 7 wt. % MgO were not used later on due to concerns about effects of fractional crystallisation. Yet, published data used as reference were not filtered in the same way and in fact, previously published data are almost entirely for samples with < 7 wt. % MgO. Those studies have argued against significant modification of Mo isotope ratios by fractional crystallisation, at least for basalts and basaltic andesites. In the Pagan data presented here, only two samples are significantly isotopically lighter than the rest and interestingly, a similar

“trend” does not exist for NW Rota. While it is clearly best to focus on primitive samples (and this study reports some of the very few Mo isotope data for high MgO samples) it seems arbitrary to selectively ignore some of the samples with lower MgO.

Please see reply to the **Overall response to reviewers (3)**.

We have re-evaluated the amphibole fractional crystallization effects on the arc basalt geochemistry. Amphibole fractional crystallization will induce increasing of Hf/Nd and decreasing of Ce/Mo and Dy/Yb of the residual magma (e.g., Tiepolo et al., 2007; Davidson et al., 2007; Voegelin et al., 2014; Wille et al., 2018). For the Izu and Mariana data cited in the text and diagrams. We can't find the amphibole fractional crystallization effects based on proxies such as Hf/Nd and Dy/Yb. We think these samples didn't experience significant fractional crystallization of amphibole, thus their Mo/Ce and Mo isotopes mostly reflect the slab dehydration and mantle melting processes.

The average eclogite from Chen et al. (2019) can reasonably represent the plate surface that experienced long-term interaction with deep fluid that is derived from serpentinite break-down in the slab lithosphere (Chen et al., 2019). Based on the available parameters, our calculations reveal melting of the average eclogite ($\delta^{98/95}\text{Mo} = -0.45\%$) from Chen et al. (2019) at 900°C can generate a melt with $\delta^{98/95}\text{Mo}$ of -0.35% , similar as the lowest $\delta^{98/95}\text{Mo}$ of the Pagan samples. For the NW Rota-1 samples except those from the East Knoll, the $\delta^{98/95}\text{Mo}$ versus Hf/Nd correlation indicate their mantle source is metasomatized by a single hydrous melt slab

component. Direct melting of the average eclogite from Chen et al. (2019) cannot get a melt with $\delta^{98/95}\text{Mo} > -0.1\%$. Our calculations reveal the average eclogite mixed with 2% fluid ($\delta^{98/95}\text{Mo}=0.29\%$) that traveled through and equilibrated with the deep crust with a fresh MORB like Mo isotope composition (original $\delta^{98/95}\text{Mo}=-0.21\%$) at 700°C can generate a new slab component with $\delta^{98/95}\text{Mo}=-0.1\%$. Melting of this new slab component can generate a slab melt with $\delta^{98/95}\text{Mo} = \sim -0.01 \%$. This is a good evidence for serpentinite breaking down to generate fluid that trigger melting of the slab surface for the deeply subducted slab. This explanation for the Mo isotopes also meets the low $^{87}\text{Sr}/^{86}\text{Sr}$ signature of the NW Rota-1 samples.

The information has been suitably added in the text.

There is no information on sample preparation in the method section. This is particularly important because the samples are from submarine eruptions and hence easily altered by seawater.

Information on sample preparation has been added in the method section. All samples were pulverized in an agate ball mill after sawing and jaw crushing.

Fig. 3A shows a much steeper trend in $d^{98}\text{Mo}$ vs. Ce/Mo for Pagan than for previously published arc sections, suggesting an isotopically heavier fluid. This is an interesting aspect of the data, in particular with regards to the mass balance of Mo in subduction zones and beyond and thus worth highlighting.

The information of the steeper trend is already highlighted in the revision. According

the $\delta^{98/95}\text{Mo}$ versus Ce/Mo correlation, the Pagan volcano mantle source may contain some slab inputs with $\delta^{98/95}\text{Mo}$ much higher than +0.2‰. This component is potentially the serpentinite formed in the mantle wedge. We have added Mo isotope data for serpentinite sample from the Asùt Tesoru mud volcano, collected during the IODP Exp. 366. These samples have $\delta^{98/95}\text{Mo}$ from 0.2‰ to 0.52‰. Please see reply to ‘**Overall response to reviewers (4)**’.

Other comments on lines:

46-47: It’s odd to introduce the fluids as derived from altered oceanic basalt and then later discussing them as derived from serpentinites. That classic AOC model is not really up to date any more.

We have rewritten part of the introduction section. Please see reply to ‘**Overall response to reviewers (1)**’.

In the revision, we first point out the problems that have been identified from radiogenic isotope studies and then introduced the importance of serpentinite (formed in forearc mantle and slab lithosphere) dehydration in completing these understandings.

136-137: Only one Pagan sample has higher Ce/Mo than the depleted mantle in Fig. 3a.

The DM has Ce/Mo of ~ 31 according the study of Salters and Stracke (2004). The $\delta^{98/95}\text{Mo}$ versus Ce/Mo correlation for the NW Rota-1 samples converge towards an

endmember of Ce/Mo= \sim 30 and $\delta^{98/95}\text{Mo}=-0.2\%$. We think 30 can reasonably represent the Ce/Mo of the DM. Then the two samples from the SW Flank of Pagan have Ce/Mo higher than the DM.

We have modified our diagrams (Figure 3a) and descriptions in the revision.

146: Radiogenic Hf and Nd in the mantle doesn't need metasomatic input.

Corrected. Radiogenic Hf and Nd isotopes are expected for melts derived from a depleted mantle source that was metasomatized by slab-derived aqueous fluids with negligible Hf-Nd contents.

179-180: Why just sediment melt? The AOC may well melt, too.

According the constraints from Hf-Nd isotopes, the slab surface is composed by AMOC and sediments with a ratios of 9:1. We use slab surface melting in the whole text of the revision. It contains contributions from both of the AMOC and sediments.

188: "To reconcile this dilemma..." I think this dilemma needs to be explained in more detail and assessed quantitatively, in particular with respect to the subsequent sentence stating that the sediment contribution is "minimal".

Please see reply to '**Overall response to reviewers (2)**'. We have added detailed calculations for the different slab components based on Mo-Nd-Hf systematics. According to the Hf-Nd isotope and Hf/Nd ratio constrains, the slab melt was derived from a mixed source consisting of subducted AMOC and sediments with a mass ratio

of 9:1 (Figures 5C and S3). Then the slab melts should have much higher $^{87}\text{Sr}/^{86}\text{Sr}$ (~ 0.705) than the low $\epsilon\text{Hf}-\epsilon\text{Nd}$ basalts (~ -0.7035). Breakdown of serpentinite in the slab lithospheric mantle may help explain these isotopic paradoxes, as it would release fluid that could travel through and equilibrate with overlying subducted oceanic crust and induce wet partial melting of the upper altered crust and sediments. This mixed hydrous melt is then delivered to the mantle source of arc magmas as a single metasomatic component (Klaver et al., 2020).

196-198: This agrees well with a similar model we have proposed for the Izu section of the arc (Freyduth et al. 2016 GCA 186 and Freyduth et al. 2019 EPSL 522).

These important early contributions based on Th-U isotopes have been cited in the revision.

Reviewer #3 (Remarks to the Author):

Review of MS NCOMMS-21-02686 “Molybdenum isotopes unmask slab dehydration and melting at Mariana arc”

Reviewed by Ivan Savov (Univ. Leeds)

This is a very well written manuscript, with excellent graphics and good data quality, including data of a novel tracer (Mo isotopes). The manuscript supports interesting and currently debated hypothesis for the mechanisms of elemental and isotope cycling from subducting plates to the surface volcanoes. The insights from the manuscript can be largely applicable to multiple and diverse disciplines of the earth and marine

sciences. With that said, I think with moderate revisions the manuscript will have high and overall positive impact and is worth publishing in a journal such as Nature Communications. **Main comments:**

The manuscript contains both new Mo isotope data, as well as published isotope and elemental datasets on these same rock samples. Some of the authors are world-leading experts on arc geochemistry and not surprisingly their sample selection is excellent. The differences with the previous Mo isotope datasets from the same arc (*but different volcanoes) are a bit worrisome, but the explanation of the authors is quite convincing and not analytical in nature. Some of the conclusions and ideas in this manuscript are in direct clash with the views of Chen et al, Nature Comm. 2019 (you have this reference on your line 390), which is also involving Mo isotopes and review of the role of serpentinites (from an Raspas, which is exhumed and well preserved ophiolite complex). In a way this is great and good for your story, making the interpretation novel. In fact your story is basically like their figure 4, but turned upside down.

Also worth noting is their AOC compositions as shown on a Mo isotopes vs Mo/Ce plot, where AOC seems to be off the trends shown by the ophiolite, which is in itself not supporting the altered crust/AOC as a “player”. I particularly agree that there are some really important insights in the correlations seen and their careful consideration does lead to the conclusions that the hydrated in the forearc and previously depleted mantle peridotites are an end member in the arc magma sources.

We thank the comments and suggestions above and below.

Chen et al., Nature Comm. 2019, Freymouth et al., 2015 EPSL, 2016 GCA and 2019 EPSL and Klaver et al., 2020 GCA all have discriminate the potential role of the slab lithosphere serpentinite in triggering slab dehydration/melting. Our Mo-Sr-Pb-Nd-Hf isotopes and elemental ratios support their earlier conclusion, i.e., deep slab serpentinite fluid triggering slab melting. In consideration of the Hf/Nd isotope variations clearly need slab melt with a contribution of sediments, the low $^{87}\text{Sr}/^{86}\text{Sr}$ of the low Hf/Nd samples thus need lithosphere serpentinite fluid to modulate the high $^{87}\text{Sr}/^{86}\text{Sr}$ signature of the subducted sediments. **In addition, based on Mo-Nd-Hf isotopes and Hf/Nd ratios we find the best and most plausible explanation for subduction inputs for the Pagan volcano is that: the source material is first metasomatized by aqueous fluids with high $\delta^{98/95}\text{Mo}$, and is then fluxed by hydrous melts with low $\delta^{98/95}\text{Mo}$, Hf/Nd, ϵNd and ϵHf .** The aqueous fluids signature is best explained by dragging down of forearc serpentinitized mantle by the slab. Please see reply to the ‘**Overall response to reviewers (2)**’.

Our key contribution of this study is that we have discriminate the different role of the two different kinds of serpentinites in subduction zone fluid. On the first hand, forearc serpentinite being dragged down by the slab may act as an important intermediate carrier for slab fluid and melt transfer in the subduction channel. On the other hand, dehydration of serpentinite in the slab lithosphere is critical for triggering slab dehydration and melting at deep depth. The breakdown of shallow fluid and deep fluid/melt fluxed serpentinite in the subduction channel triggers Mariana volcanic front volcanism. The two kinds of serpentinites are a partnership of coexistence, not a

rivalry of one or the other. Please see reply to the ‘**Overall response to reviewers (1)**’.

We have re-written part of the introduction and text to clarify our main finding.

This has been suggested more than a decade ago via trace element arguments, with Pb-Nd-B isotopes (Tera and co-workers, Ishikawa-san & Nakamura-san; Ryan and co-workers, among others). Adding another tracer for support of the forearc mantle contribution to arc magmas is outstanding achievement! It is novel in that the data is extending the variations of arc volcanic rocks previously reported and via combination of FME/Nb ratios it is convincingly supporting a widely debated issue of the type of serpentine input into the arc magmatic source- lithospheric in origin (deep MORB mantle at bottom of gabbroic slabs) or forearc modified and down dragged with the slab to depths and with ultimately (ultra)-depleted mantle protoliths. I urge the authors to browse through the recently published paper (in Nature Comm.) that reports on the modelling of fluid penetration in deep slabs to form serpentinites and the d11B signatures of the altered oceanic crust (that is eventually subducted). This study (reference is below) is another independent evidence for lack of arc contributions from the hydrated lithospheric section of the slabs. This fact is leaving the forearc fluid-modified mantle as the only other viable alternative (and more reasonable to be honest as hydrous slabs will be hard to subduct due to low density!).

McCaig, A.M., et al. No significant boron in the hydrated mantle of most subducting slabs. Nature Comm. 9, 4602 (2018). <https://doi.org/10.1038/s41467-018-07064-6>

The important constraints from the study of FMEs and boron isotopes are already reviewed in the introduction section. Please see reply to the '**Overall response to reviewers (1)**'.

We agree with the conclusion of McCaig et al. (2018) that the lithosphere serpentinite may be not enriched of boron. It is consistent with the oceanic drilling data that lower crust gabbro has low boron content (Smith et al., 1995). However, the serpentinite breakdown released fluid are important for extraction of Mo-Pb-Sr from the lithosphere.

The conclusions of (this) manuscript are also similar to the study of Kimura-san, where there is fantastic and very quantitative estimates of the nature of inputs from the slab and mantle under cross arc volcanic chains across the Izu arc (fluid X= 2-4% shallowly and meltX=1.5-4.5% for the deep melts). Their conclusions were derived from trace elements and also from Sr and Pb isotopes in combination with trace elements. All such data is apparently available for the mafic rocks from the Marianas (this study), so some links perhaps can be further established. I accent on the idea of the manuscript to be a bit more quantitative. I suggest that the latest version of arc basalt simulator (ABS) can used to show some parallels with the Kimura-san's 2010 study (doi:10.1029/2010GC003050) . Perhaps there may be some bridges that can prove useful and importantly some quantifications from the Marianas may be revealing a more global case.

We thank the suggestion for making mass balance calculation using the ABS. We have tried to make the calculation using the latest version, ABS5 (Kimura et al., 2017). This software is designed to calculate the total subduction inputs. Given that we've identified different slab components, shallow slab fluids, deep slab melts, and slab lithosphere serpentinite fluids, we chose to perform manual calculations using available parameters, which allowed us to calculate Mo and Mo isotopes of slab melt/fluid and the serpentinite fluid, a function not yet available for ABS5.

Please see reply to the '**Overall response to reviewers (2)**' for details.

Your figures 3-bottom two segments with Ba/Nb and Cs/Nb- here one may argue that there are two trends, but those trends are not necessarily the one you highlight. If we want to include the Izu arc in the discussion, it will be easier to have trend 1 [steep trend] consisting of all NWRota-Pagan (COB 2)-the high Ba/Nb samples from Alamagan and Guguan-all of the Izu arc data (with a mixing end member DM); and trend 2 (more vertical one) = including Agrigan and Uracas volcanoes-NW Pagan and the highest 98/95 Mo samples from Alamagan and Guguan volcanoes.

We thank the suggestion for other grouping strategy. The suggested classification fit the $\delta^{98/95}\text{Mo}$ versus Mo/Nb, Ba/Nb, Cs/Nb correlations well, but not for the $\delta^{98/95}\text{Mo}$ versus Ce/Mo correlation (Figure 3-a), and not for the $\delta^{98/95}\text{Mo}$ and Ce/Mo versus Hf/Nd correlations (Figures 4e). Our grouping are based on sample localities, i.e., different volcanoes. But we added information that different $\delta^{98/95}\text{Mo}$ versus Mo/Nb, Ba/Nb, Cs/Nb correlations may be affected by the different degree of mantle depletion.

This is consistent with the different $\delta^{98/95}\text{Mo}$ versus ϵHf and ϵNd correlations (Fig. 5).

I urge the authors to take the forearc serpentinitized mantle trace elements from, say Savov et al., 2007-JGR and see where on the trace element graphs these potential end-member compositions will/may plot. Site 801C is clearly not very useful to explain the mixing relationships and so adding something in the high Ba/Nb and Cs/Nb end of the plot and always with very high 98/95 Mo will be quite useful. What is in the upper right corner of your diagrams, anyway? Why is Izu arc and the NE flank of Pagan trending in this direction? Please also note that there may be two different trends on your figure 4 (Mo isotopes vs. Hf isotopes). What is sitting in the upper right corner there? Please add an end-member on the plot. Again the trends here do not go through 801C basalt composite. I suppose that some of the highly modified serpentinites will have elevated 98/95Mo (aqueous fluid arrows on the other plots show vertical enrichments of 98/95Mo) and so these may be good to show (or if anyone have looked at serpentinites for Mo isotopes- those will be good to see as they are central to your hypothesis).

The endmember in the upper right corner should be the mantle wedge serpentinite dragged down by the slab from a depth shallower than 80km. The Asùt Tesoru mud volcano is much closer to the Pagan volcano than other serpentinite mud volcanos with data reported. We analyzed Mo isotopes for some samples from the Asùt Tesoru mud volcano during the review process as part of a collaboration with co-author Ryan, and the data are now plotted in the Fig. 3. As you supposed they have high $\delta^{98/95}\text{Mo}$.

Figure D1. Ba/Th versus Th diagram for the serpentinites. The data of Yinazao, Fantangisna and

Asut Tesoru are unpublished data of Li and Ryan et al.

The samples from Asut Tesoru mud volcano have Th-Hf-Zr-Ti contents being much higher than samples from the South Chamorro (data from Savov et al 2007), indicating the protolith of the Asut Tesoru serpentinite may have experienced a lesser degree of earlier melt extraction than those for the South Chamorro seamounts. The Cs/Nb and Ba/Nb are much lower in the Asut Tesoru samples than those in the South Chamorro. Assume all the serpentinites from different seamounts have experienced metasomatism by a fluid with similar Cs-Ba, then the Cs/Nb and Ba/Nb are also largely affected by their original Nb in the peridotite protolith. As the Nb of the Asut Tesoru samples are generally two orders of magnitude higher than samples from the South Chamorro seamount, their Cs/Nb and Ba/Nb are reasonably two orders of magnitude lower in these samples.

We should point out that these serpentinites will be further metasomatized by the slab fluid when they are dragged down by the slab to a depth of > 80 km, where the slab starts to couple with the overriding mantle and being heated up to devolatilize.

The mantle wedge serpentinite we discriminated is not exactly equivalent to serpentine mud volcanoes that erupt at the forearc corresponding to a very shallow plate depth (< 19 km).

There is now published dataset for adakitic melts, effects of hornblende in the source of melts and Mo isotopes. The work is published in *Geochim Cosmochim. Acta* (<https://doi.org/10.1016/j.gca.2021.01.020>). It appears to show that hornblende crystallization may dominate the Mo isotope variations in arc magmas. Please consider this new dataset, especially when you discuss the deep sourced magmas. There may be useful information about the effect of mineralogy and mantle metasomatism. Also please check if there is anything in the literature on Mo isotopes in metasomatized/serpentinized rocks.

Using their K/Rb proxy to monitor Am fractionation effect get the same conclusion based on Dy/Yb and Hf/Nd indicators (Davidson et al., 2007; Tiepolo et al., 2007). This reference is cited in the revision.

We added Mo and Mo isotope data for serpentinite samples. Please see reply to the ‘**Overall response to reviewers (2)**’ for details.

There are small details that I think may be helpful to clarity, especially for the non-expert (petrologists and arc geochemist) audience of Nature Comm. I list below some of those points and some of the text edits that will help.

Minor details linked to the text:

line 39- will help if you state “intraoceanic”. The mentioning of end-member convergent margin is not helpful. I rather add that it is “non accreting” and also add that it is quite sediment starved, making it appropriate, if not unique for the cross arc cycling studies.

The sentences have been re-written in the revision:

The Izu-Bonin-Mariana (IBM) arc system stretches over 2,800 km from near Tokyo, Japan to south of Guam, USA, and is a typical intraoceanic island arc with negligible inputs from the sub-arc crust to the lavas. The IBM system is an endmember non-accreting convergent margin with a thin sedimentary cover on the downgoing slab, which means that input and output fluxes in the subduction zone may be more confidently assessed (Stern et al., 2004).

line 47- no CAPS for Boron

Corrected

line 48- The cited studies does not involve Hf isotopes. Perhaps the study of J.Pearce on the Hf isotopes of the Marianas will be good to add.

Rewording:

The aqueous fluid component has been presumed to derive from subducted altered mafic oceanic crust (AMOC) based on heavier boron isotopes and more radiogenic Nd isotopes, indicating a less sediment affected mantle source (Elliott et al., 1997; Ishikawa and Tera, 1999). The hydrous melt component is from subducted sediments,

given its high Th/Nb and less radiogenic Nd isotopes (Elliott et al., 1997).

line 50- Please note that the Nd and Hf are NOT a good tracers of fluids as the elements in question are highly fluid immobile. In the case of Nd- please note that there is increasing amount of evidence in the literature that the $^{143}/^{144}\text{Nd}$ of serpentinites may indeed vary vastly (research by Bizimis and co-workers) and the process behind it is not well understood.

Please see reply above. Here we introduce the problems of study based on other isotopes and trace element ratios.

line 53- “inconsistent with the Mariana arc lavas”- perhaps add the range here.

The range is added (0.7031-0.7041; Straub, 2017).

Line 54- these are indeed some moderately elevated $^{87}/^{86}\text{Sr}$ values, but this is in respect to MORBs. Otherwise, in respect to anything in the slab those are immensely low or better- unradiogenic.

We select to use the word “unradiogenic”.

line 74-77- what are the errors for the Mo isotope ratios. If those are large, then the MORBs and sediments may nearly overlap. This is critical point why we need AOC as end-member and why you have shown exactly that on your plots (ODP Site 801 end member)

The range for different slab components have been added.

AMOC (-0.12‰ to +0.86‰, with a weighted average of +0.36‰) and marine sediments (-1.87‰ to +0.11‰, with a weighted average of -0.31‰; Freymuth et al., 2015) have $\delta^{98/95}\text{Mo}$ that are, respectively, higher and lower than the depleted mantle (DM; $\delta^{98/95}\text{Mo} = -0.21 \pm 0.02\%$; Bezard et al., 2016).

Line 84- You need to state how we know these depths. What methods were used to determine where the slabs are and at what dip they sink in the mantle. A reference will also be good.

The depth is calculated based on the sample position and slab position in the mantle (Hayes et al. 2018; Slab2 data).

line 108- East Knoll is with CAPS.

Corrected.

line 112- you may want to state somewhere what is the slab DIP. One way to do this is maybe in the caption of your schematic summary diagram. In any case- for non specialists there is a need to explain that. Another option will be to add it as a method.

Method added in the 'introduction section'. The depth is calculated based on the sample position and slab position in the mantle (Hayes et al. 2018; Slab2 data).

line 125- but this is not too distant range I respect to the Izu VF (0.16 per mil) , isn't it?

This is one of the reasons that the range of the isotope errors needs to be properly reported.

Rewording:

The NW Rota-1 samples are lower in $\delta^{98/95}\text{Mo}$, from -0.20‰ to -0.02‰, and show a smaller $\delta^{98/95}\text{Mo}$ range ($0.18\text{‰}\pm 0.12\text{‰}$) than either the Pagan samples ($0.4\text{‰}\pm 0.12\text{‰}$) or other IBM samples ($0.32\text{‰}\pm 0.12\text{‰}$).

Line 132- Cs/Nb are high in fluid, no doubt. This has been shown nicely in Savov et al, 2007-JGR manuscript.

Reference cited.

Line 140- I recommend that for the sake of consistency you stick to your Ce/Mo (as in your line 126) and do not confuse things by introducing Mo/Ce. So here you may say “ high Ce/Mo”.

We use Ce/Mo consistently in all the text and diagrams in the revision.

line 168- elevated Sr isotope ratios in respect to MORB. And only so very little!

Unradiogenic $^{87}\text{Sr}/^{86}\text{Sr}$. Corrected.

line 171-172- Need to tell us if this fluid is realistic (see my recommendation for use of McCaig et al, 2019) . In any case- if this fluid exists [doubt that!] is released- it will react with the abundant fresh gabbro and diabase and will not manage to do anything.

If you eliminate this option early- then you can use some lines for properly introducing the forearc down dragging processes (which are not vague) and link to the Mariana md volcanoes, which are physical evidence for the high Ba and Ca you need. Please see reply to the '**Overall response to reviewers (1)**'. Our conclusion is that: both of the serpentinite formed in the forearc mantle and slab mantle are important for subduction zone fluid generation.

Brucite in the slab serpentinite will breakdown at temperature 300-400□ (Plümper et al., 2016; Peters et al., 2020). According the study of Erro-Tobbio meta-serpentinites (Ligurian Alps, Italy), the fluid can escape via channel networks from the slab mantle. The reaction is: Antigorite + Brucite= 2 Olivine + 3 H₂O (Plümper et al., 2016; Peters et al., 2020). According the thermal structure of the Mariana arc, the slab surface has much higher temperature than the slab Moho depth (Syracuse et al., 2010), the slab serpentinite fluid may be heated up when migrating up through the crust and incubate the ability to extract the Pb-Sr-Mo elements from the crust.

This place we are discussing the fluid components (high Sr/Nd, Cs/Nb and Ba/Nb component) of the arc lavas. Although this component has Sr/Nd≈ 50, being much higher than that for the DM (~ 10). This component have unradiogenic ⁸⁷Sr/⁸⁶Sr and DM like $\delta^{88/86}\text{Sr}_{\text{SRM987}}(\text{‰})$ (Fig. 9 in Klaver et al. 2020 GCA). The ⁸⁷Sr/⁸⁶Sr and $\delta^{88/86}\text{Sr}_{\text{SRM987}}(\text{‰})$ are not affected by isotope fractionation during slab dehydration. In addition, the Th-U-Pb isotope systems also indicate this component have fresh MORB like isotope compositions (Freymuth et al., 2015, 2016, 2019). These isotopes need

the fluid (generated by slab serpentinite breakdown) composition to travel through and equilibrate with the overlying oceanic crust to modulate the isotopes to that of pristine MORB. On the one hand we think the mantle wedge serpentinite will be dragged down, on the other hand we think they will be further metasomatized by the slab fluids. The slab fluid may be related to breakdown of the plate serpentinite.

We did some rewording in the revision to clarify our discussion.

Line 180- why is this. Maybe clarify your arguments.

Please see reply to the '**Overall response to reviewers (2)**'.

Rewording: This component likely reflects slab melt component with 10% contribution from the sediments.

line 184- this increasingly make more and more sense to me.

line 193 (and several other places in the manuscript)- a just published GCA paper by Anders et al (2021) is reporting very interesting story from Izu arc and is telling us that sediments are not playing a role and that the slabs are melting. Please add this paper insights (Sr, Pb, Nd, Hf isotopes are VERY abundant!) to your story. I think it is highly relevant to see what is the entire range of mantle melts, nicely shown vs. sediments- all nicely summarized in their plots.

We thank the suggestion for quantitative calculation based on radiogenic isotopes. We have added calculation based on Hf-Nd isotopes and Hf/Nd ratios. Please see reply to the '**Overall response to reviewers (2)**'. Our calculation result of slab met with ~ 10%

contribution from the sediment is consistent with the results from McCarthy et al. (2021 GCA). The reference paper is added.

line 196- need reference here after t”sediments”

The references added after next sentence.

line 229- “from in”

‘in’ is deleted

line 232- hence the two trends on figure 4B (Hf isotopes)

Added: hence there are two trends on $\delta^{98/95}$ Mo versus ϵ Nd and ϵ Hf, and ϵ Hf versus ϵ Nd correlations (Fig. 5)

line 236- This is a good place to cite some forearc serpentinite peridotite major and trace element paper.

Some important references have been cited here and in the introduction section.

line 239- it will be more thorough if here you also cite McCaig et al. (2019).

cited

Line 246- here refer to some of your figures

Figure 4 and Figure 5 are refereed here

line 254-this “dragging down” of the serpentinitized mantle needs to be either explained in a bit more detail or some key references need to be given. As it is, for average reader, this is not clear enough.

Please see reply to the '**Overall response to reviewers (2)**'. We added detailed calculation based on Hf-Nd-Mo systematics and the important references have been cited.

line 255- this is SUPER! I really think this is a great selling point of your paper and you may want to further accent on this fact, in addition to the mixing trends and arc magma genesis. People should start plotting this as mixing end member and not some composite samples of AOC or sediments, which may but may not be relevant.

Please see reply to the '**Overall response to reviewers (2)**'. We added detailed calculation based on Hf-Nd-Mo systematics to clarify our argument on dragging down of forearc serpentinite.

line 263- "Breakdown of serpentine ..."- see McCaig et al. (2019). Also note than at high T this breakdown will lead to formation of chlorite- rich protoliths.

The fluid-rock interaction is affected by the thermal structure of the slab, therefore, fluids may behave differently on the way down and up.

Brucite in the slab serpentinite will breakdown at temperature 300-400°C (Plümper et al., 2016; Peters et al., 2020). According the study of Erro-Tobbio meta-serpentinites (Ligurian Alps, Italy), the fluid can escape via channel networks from the slab mantle. The reaction is: Antigorite + Brucite= 2 Olivine + 3 H₂O (Plümper et al., 2016; Peters et al., 2020). According the thermal structure of the Mariana arc, the slab surface has much higher temperature than the slab Moho depth

(Syracuse et al., 2010), the slab serpentinite fluid may be heated up when migrating up through the crust and incubate the ability to extract the Pb-Sr-Mo elements from the crust.

The alteration temperature of the Hess Deep gabbro is only about 200 °C (McCaig et al., 2019).

Line 272- please give reference here for the depth.

This depth is estimated according our Mo/Ce and Mo isotopes, i.e., a high Mo isotope low Ce/Mo component disappeared for the NW Rota-1 samples.

Just a note to the authors, that serpentinites often have Ba/Th (10^3 and 10^4).

Please see reply to the main concern. The Ba/Th is also significantly affected by the degree of mantle depletion for the serpentinite. As the Th for samples from the South Chamorro can be 2-3 orders of magnitude lower than those from the Asùt Tesoru samples, their Cs/Th are reasonably 2-3 orders of magnitude higher than the later. We think the depletion of Th may be related to early melt extraction from the forearc mantle, either during subduction initiation or more ancient event (Li et al., 2019a; Parkinson et al., 1998). Different degrees of forearc mantle depletion may also affect the corresponding volcanic arc basalt chemistry, e.g., the high Ba/Nb and Cs/Nb signature of the Izu volcanic front samples (Figure 3).

Reviewers' Comments:

Reviewer #1:

Remarks to the Author:

In the revised version of this manuscript as well as in their detailed replies, the authors have adequately addressed my concerns. Now this work will be a valuable contribution that I recommend for publication in Nature Communications in its present form.

Reviewer #2:

Remarks to the Author:

The paper has improved during revisions and is well written and presented. Data for seamount samples were also added which are interesting.

But the way the seamount samples are presented could be improved. Despite there being only five datapoints, the data are shown as rectangular fields in Fig. 3 which are quite large and make it impossible to judge where the actual data are located in the plots. I suggest that these are replaced by the actual data instead.

My main point of criticism is that I believe the importance of the forearc serpentinites in generating the arc magma chemical inventory is overstated:

- The main argument against a model that could explain the data without the involvement of forearc serpentinites is discussed in lines 273-279, namely that the temperature variation corresponding to varying slab depths beneath Pagan is too low to explain significant variation in slab melt + slab fluid mixtures. This is a rather surprising argument, as – if I understand it correctly – it assumes that fluids and melts generated beneath the arc move towards the surface strictly vertically. There is a lot of evidence that fluid and melt migration pathways are highly complex and some studies explicitly suggest lateral flow and that magmas might frequently tap into backarc sources to produce geochemical variation (see e.g. Ishizuka et al. 2015, EPSL).

- Very similar geochemical models have been presented that do not require forearc serpentinites as a source for Mo, see Villalobos-Orchard et al. 2020 whose models can be directly compared to those in Fig. 5.

In conclusion, I believe that there is a possibility that forearc serpentinites are involved in arc magma formation and this is well worth pointing out, as done in this manuscript. Yet, the same data can be explained without such a model and the new data are by no means a smoking gun for the involvement of fore-arc serpentinites. It could be presented as the author's preferred model but unless a stronger case can be made I think it is not appropriate to present it as 'requirement'.

There seem to be some inconsistencies in the models. In particular, the need for the deep slab melt (beneath NW Rota) to have higher $d_{98}\text{Mo}$ than the shallower slab melt beneath Pagan is odd, as the slab loses isotopically heavy Mo at shallower levels. From the text, it seems that the 6 GPa melt is actually a mixture of a melt and a fluid, while at 4 GPa the fluid and melt components are treated separately. This needs some clarification.

Can the geochemical models be added to the $d_{98}\text{Mo}$ vs. Ce/Mo plot (or a panel added to Fig.5)? All the parameters seem to be available. It would be interesting to see if the models fit the data in that plot.

Lines 169-172: The 'much steeper trend' is impossible to see in Figure 3a because trendlines in that figure should not be linear if one of the axes is logarithmic.

Lines 379-381: The argument that no mixing trends are seen with the depleted mantle (DM) is not good here (see also my previous review). Even the geochemical models in Fig. 5 include the DM, yet

there are no mixing trends seen with it either.

Figure 3a: Mixing trends are not linear in this figure when one of the axis is logarithmic. Hence, it is not sensible to plot linear trendlines. I suggest to use a linear x-axis instead.

Heye Freymuth, hf325@cam.ac.uk

Reviewer #3:

Remarks to the Author:

Dear Hongyan Li and co-authors. I thank you for your detailed descriptions and alterations of the text and figures as I and the other reviewers suggested. I was particularly happy to see that you have added Mo dataset for Asut Tessori seamount and that you find this effort useful. The new figure and the impact of serpentinized forearc mantle rocks is now clear in my view. The figures have massively improved and their impact and link to the proposed novel sources is better defined and hopefully much sharper. I find your story fascinating and although not all of the NG audience understands your tracers, I think you have done the best possible effort to convince us. With that said- I don't have more comments and now recommend this manuscript for publication.

Good luck!

Ivan Savov (Leeds Univ.)

DETAILED RESPONSES TO THE REVIEWERS

Reviewer #2 (Remarks to the Author):

The paper has improved during revisions and is well written and presented. Data for seamount samples were also added which are interesting.

But the way the seamount samples are presented could be improved. Despite there being only five data points, the data are shown as rectangular fields in Fig. 3 which are quite large and make it impossible to judge where the actual data are located in the plots. I suggest that these are replaced by the actual data instead.

Thanks for the suggestion. The serpentinite data have been presented with the actual data points in the new Fig. 3. We have also adjusted the y-axis to show all the relevant data as possible.

My main point of criticism is that I believe the importance of the forearc serpentinites in generating the arc magma chemical inventory is overstated:

- The main argument against a model that could explain the data without the involvement of forearc serpentinites is discussed in lines 273-279, namely that the temperature variation corresponding to varying slab depths beneath Pagan is too low to explain significant variation in slab melt + slab fluid mixtures. This is a rather surprising argument, as – if I understand it correctly – it assumes that fluids and melts generated beneath the arc move towards the surface strictly vertically. There is a lot of evidence that fluid and melt migration pathways are highly complex and some studies explicitly suggest lateral flow and that magmas might frequently tap into back arc

sources to produce geochemical variation (see e.g. Ishizuka et al. 2015, EPSL).

- Very similar geochemical models have been presented that do not require forearc serpentinites as a source for Mo, see Villalobos-Orchard et al. 2020 whose models can be directly compared to those in Fig. 5.

In conclusion, I believe that there is a possibility that forearc serpentinites are involved in arc magma formation and this is well worth pointing out, as done in this manuscript. Yet, the same data can be explained without such a model and the new data are by no means a smoking gun for the involvement of fore-arc serpentinites. It could be presented as the author's preferred model but unless a stronger case can be made I think it is not appropriate to present it as 'requirement'.

We thank these constructive suggestions and accept the criticisms. We have added some words to indicate that this is our preferred model, which need to be checked by future research.

Ishizuka et al. (2015, EPSL 430: 19-29) presented a good example for addition of Izu-Tobu (backarc) magma to the Izu-Oshima (volcanic front) magma plumbing system, while addition of Izu-Oshima magma to the Izu-Tobu is not observed. On a larger temporal-spatial scale, the Mariana and Izu arcs may be different in terms of transporting of slab material in the mantle wedge as a result of different mantle convection (e.g., with or without back-arc spreading). A model that rear-arc mantle metasomatised by deep-sourced slab melt, and then further metasomatised by cooler, H₂O-rich fluids near the arc volcanic front, as proposed by Hochstaedter et al. (2001, G3; <https://doi.org/10.1029/2000GC000105>) for the Izu-Bonin arc, maybe another

factor affecting the geochemistry of the Izu-Bonin arc magmatism.

Figure S4. *Th/Nb versus Pb/Ce diagrams for Pagan and NW Rota-1 samples unaffected by amphibole fractional crystallization. Izu-Oshima and Izu-Tobu samples (Ishizuka et al., 2015) are also plotted for comparison in diagram (b). The variation of the Izu-Oshima (volcanic front) samples is explained by addition of Izu-Tobu (backarc) magma to the Izu-Oshima magma plumbing system. This model is difficult to explain the geochemistry of Pagan samples. Note the high Pb/Ce and high Th/Nb characteristics indicate the slab fluid for the Oshima volcano may be supercritical. It is different with that for the Pagan volcano.*

Although the role of mixing of slab components in the mantle wedge or mixing of magmas in the plumbing system cannot be fully ruled out (Ishizuka et al., 2015), this does not explain the Th/Nb versus Pb/Ce variation in the Pagan samples (Fig. S4). Our $\delta^{98/95}\text{Mo}$ versus Ce/Mo plot (Fig. 3a) also doesn't support a model of mixing magmas from the rear arc to the volcanic front. The phenomena that the fluid signature gradually decreased while the melt signature gradually increased along with the increasing of slab depth best fit the Pagan data of this study (Fig. S4). Covariations between fluid proxies and melt proxies indicate the two slab components were possibly first captured by a source contributor before being released to the magma mantle source.

We have indicated that the forearc serpentinites added to the Pagan volcano source is not exactly the same as the Asùt Tesoru serpentinites. The fluid component for Pagan volcano bears similarities to the Asùt Tesoru serpentinites in that they have high $\delta^{98/95}\text{Mo}$, low Ce/Mo and high Mo/Nb. However, the high Cs/Nb and Ba/Nb signature of the volcanic lavas (Figure 3) are difficult to explain through serpentinite inputs alone, as Cs is only moderately (25% to 30 %) while Ba is only slightly (< 2%) mobile off the subducting slab at 10-40 km depths (Savov et al., 2005, 2007). Further metasomatism of the subduction channel material by slab fluids released at depths > 40 km is necessary, under slab thermal conditions hot enough to mobilize Ba.

Other more fluid sensitive isotope systematics, like boron, can be used to test our model (Benton et al., 2001; Savov et al., 2005, 2007; Pabst et al., 2012). Detailed geophysical observations may also be helpful.

There seem to be some inconsistencies in the models. In particular, the need for the deep slab melt (beneath NW Rota) to have higher $\delta^{98}\text{Mo}$ than the shallower slab melt beneath Pagan is odd, as the slab loses isotopically heavy Mo at shallower levels. From the text, it seems that the 6 GPa melt is actually a mixture of a melt and a fluid, while at 4 GPa the fluid and melt components are treated separately. This needs some clarification.

According to the $\delta^{98/95}\text{Mo}$ vs. Ce/Mo diagram (Fig. 3a), it is clear the hydrous melt for the NW Rota-1 is not a mixture of fluid and melt for the Pagan. As the slab would lose isotopically heavy Mo at shallower levels, the heavy Mo and high Ce/Mo signature of

NW Rota-1 need the deeply subducted crust to be fluxed with high $\delta^{98/95}\text{Mo}$ fluid. This phenomena is first observed from this study according our knowledge. This fluid may be from the deep lithosphere that has been rinsed by early fluid (experienced early Mo loss) at shallow depth. We have improved a little bit of the calculations to better depict the slab dehydration and melting process based Ce/Mo and $\delta^{98/95}\text{Mo}$. Major improvements include:

1) The eclogite Ce/Mo and $\delta^{98/95}\text{Mo}$ is re-examined. Average composition of eclogite and blueschist from Chen et al. (2019) is selected. Two abnormal data (Samples SEC43-1 and SEC43-1) according the $\delta^{98/95}\text{Mo}$ versus Ce/Mo correlation are not included.

2) Melting of the slab with the average eclogite Ce/Mo and $\delta^{98/95}\text{Mo}$ composition can generate a melt like the SW Pagan samples (Table S1). **It indicates a deep lithosphere fluid for melting of the slab beneath the SW Pagan is not necessary** (Fig. S5). The Sr-Pb isotope signatures of the slab may have been buffered toward the MORB values as a result of long-term early lithosphere fluid percolation at shallow depth.

3) The slab at 6 GPa is assumed to have experienced early melting at 4 GPa (900 °C; F=10%). Deep lithosphere fluid at 6 GPa is assumed to equilibrate with a MORB-like source that has experienced 2% fluid percolation and Mo loss at shallow depth. Then melting of a mixture of 98% slab + 2% deep fluid at 6 GPa can generate the Ce/Mo and $\delta^{98/95}\text{Mo}$ signature of the NW Rota-1.

These calculations suggest that the dehydration of the slab lithosphere

serpentinite may be episodic. At slab depth < 170 km, the dehydration is controlled by mineral reaction of Antigorite + Brucite = 2 Olivine + 3 H₂O at low temperature (> 300 °C; Plümpner et al., 2016; Peters et al., 2020). According the slab Moho temperature of the south Mariana arc, the reaction between antigorite and brucite can start at ~ 100 km slab depth (Fig. D1). At slab depth > 200km, the slab Moho can reach a temperature of > 550 °C. Then dehydration of the lithosphere is possibly controlled by breakdown of antigorite (Ulmer and Trommsdorff, 1995).

These information has been added in the text.

Figure D1. Depth-temperature path of the slab Moho for the south Mariana arc according the data of D80 model from Syracuse et al. (2010).

Can the geochemical models be added to the d₉₈Mo vs. Ce/Mo plot (or a panel added to Fig.5)? All the parameters seem to be available. It would be interesting to see if the models fit the data in that plot.

Figure S5. $\delta^{98/95}\text{Mo}$ versus Ce/Mo diagrams for the Pagan and NW Rota-1 samples unaffected by amphibole fractional crystallization, illustrating the slab dehydration/melting process. The depleted mantle (DM), 700 °C slab fluid, 4 GPa slab melt and 6 GPa slab melt compositions are listed in the Supplementary Table 1. The numbers on the mixing curves between different compositions represent the mass percentage of the slab fluid/melt. Blue line in (a) is the mixing trend between DM and the 700 °C slab fluid. Pink line in (b) is the mixing trend between the partially serpentinized mantle (DM + 1% 700 °C fluid) and the 4 GPa slab melt. Gray line in (b) is the mixing trend between the DM and the 6 GPa slab melt.

We have added $\delta^{98/95}\text{Mo}$ versus Ce/Mo diagram (Fig. S5) in the supplementary diagram. The pink line in (b) is presented to show the slab components for the SW Pagan samples. DM + 1% 700 °C fluid + ~ 10% 4 GPa slab melt best fit the $\delta^{98/95}\text{Mo}$ and Ce/Mo of the SW Pagan samples. It is similar as the constraints from $\delta^{98/95}\text{Mo}$ versus Hf/Nd correlation (Fig. 5d): DM + ~ 1% 700 °C fluid + ~ 9% 4 GPa slab melt. The information has been added in the text.

Lines 169-172: The ‘much steeper trend’ is impossible to see in Figure 3a because trendlines in that figure should not be linear if one of the axes is logarithmic.

The new Fig. 3a is changed to be a linear-linear correlation. Now, the 'much steeper trend' is clear.

Lines 379-381: The argument that no mixing trends are seen with the depleted mantle (DM) is not good here (see also my previous review). Even the geochemical models in Fig. 5 include the DM, yet there are no mixing trends seen with it either.

Thanks for pointing out this. This sentence has been deleted from the text.

Figure 3a: Mixing trends are not linear in this figure when one of the axis is logarithmic. Hence, it is not sensible to plot linear trendlines. I suggest to use a linear x-axis instead.

We have accordingly changed the new Fig. 3a to be a linear-linear correlation.

Reviewers' Comments:

Reviewer #2:

Remarks to the Author:

I appreciate the added detail to the model calculations and changes made during revisions to improve the models.

I only have a few minor comments which I think the authors should be able to address without the need for additional reviews:

- The slab melt d98Mo (note p in, table S1) is calculated for a rather high temperature (1175C), much higher than the input temperatures used for the other parameters (e.g. 900C, o,r in table S1). Could a lower temperature be used so that the models are internally consistent? Also, (p) suggests that the slab melts at 4 GPa and 6 GPa have been calculated in the same way which I think it not quite right because a deep fluid was added at 6 GPa.

Lines 351-352: "Our calculations show that melting (F=10%) of the upper oceanic crust (original 98/95 Mo=-0.9‰) ... " Where does the number -0.9 come from? It seems very low!

Supplement:

S63-63 "The deep lithosphere fluid is assumed to equilibrate with a MORB-like source..." - I think it should rather equilibrate with an eclogite source.

DETAILED RESPONSES TO THE REVIEWERS

Reviewer #2 (Remarks to the Author):

I appreciate the added detail to the model calculations and changes made during revisions to improve the models. I only have a few minor comments which I think the authors should be able to address without the need for additional reviews:

- The slab melt $\delta^{98}\text{Mo}$ (note p in, table S1) is calculated for a rather high temperature (1175C), much higher than the input temperatures used for the other parameters (e.g. 900C, o,r in table S1). Could a lower temperature be used so that the models are internally consistent? Also, (p) suggests that the slab melts at 4 GPa and 6 GPa have been calculated in the same way which I think it not quite right because a deep fluid was added at 6 GPa.

We re-arranged the notes to further clarify our calculations.

^o Mo isotopes calculated according Mo isotope equilibrium fractionation factor of $\Delta^{98/95}\text{Mo}_{\text{melt-rutile}} = 0.5 \text{ ‰}$ at 900°C between the melt and the residual rutile.

^p Mo isotopes calculated according Mo isotope equilibrium fractionation factor of $\Delta^{98/95}\text{Mo}_{\text{fluid-rutile}} = 0.73 \text{ ‰}$ at 700°C between the fluid and the residual rutile.

^q Mo isotopes calculated according Mo isotope equilibrium fractionation factor of $\Delta^{98/95}\text{Mo}_{\text{melt-rutile}} = 0.5 \text{ ‰}$ at 900°C between the melt and the residual rutile.

All these fractionation factors are calculated according the experimental results of $\Delta^{98/95}\text{Mo}_{\text{melt-rutile}} = 0.33 \pm 0.06\text{‰}$ at 1175°C between the melt and the residual rutile (Chen et al., 2019). This information has been added in the text (line 275-277) and notes (o, p and q) for Supplementary Table 1.

After re-arrange the notes, now it is clear how the melts at 6 GPa is calculated.

Note q: Trace element contents calculated for model batch melting of deep fluid fluxed slab (2% Deep fluid+98% slab) at 900°C with F=10%. The slab is assumed to have experienced early melting at 4 GPa (900°C; F=10%).

Lines 351-352: “Our calculations show that melting (F=10%) of the upper oceanic crust (original 98/95 Mo=-0.9‰) ... “ Where does the number -0.9 come from? It seems very low!

The number -0.9‰ is the calculated $\delta^{98/95}\text{Mo}$ of the residue slab after melting (F=10%) at 130 km (4 GPa). The information is added in the text (Line 324-325).

Supplement:

S63-63 “The deep lithosphere fluid is assumed to equilibrate with a MORB-like source...” - I think it should rather equilibrate with an eclogite source.

Corrected.